# ADAPTIVE ROUTING OF EXPERTS VIA FUZZY RULE INTERPOLATION FOR EFFICIENT MULTIMODAL LEARNING

## ABSTRACT

Multimodal learning integrates heterogeneous data such as text, images, and audio, but existing Mixture of Experts (MoE) frameworks still face two critical limitations: (1) fixed skip rates that fail to adapt to task difficulty, and (2) inefficient expert selection guided by static priors. These issues hinder scalability and stability in large-scale scenarios with diverse semantic patterns. To address these limitations, we propose a **Fuzzy Router** that incorporates fuzzy rule interpolation (FRI) into Routing of Experts (RoE) for adaptive and efficient expert selection. A sparse fuzzy rule base, derived from prior knowledge and expert experience, is expanded via interpolation to enable nuanced routing decisions based on task complexity. This design dynamically adjusts skip rates, reduces redundant computation, and maintains accuracy through adaptive refinement. Experiments across multiple multimodal benchmarks demonstrate that our method reduces routing data and time by more than 68%, shortens overall training time by 16.7%, and preserves competitive performance. These results highlight FRI as a principled mechanism for adaptive resource allocation, advancing efficient and scalable multimodal MoE/RoE systems. Our code is open-sourced in the supplementary materials.

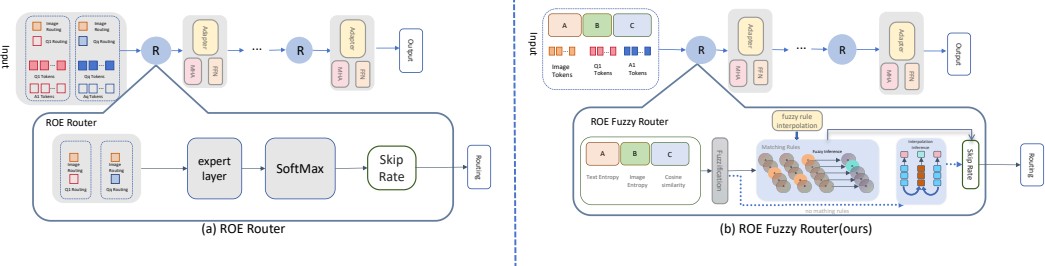

Figure 1: Comparison of routing mechanisms. (a) Baseline ROE Router employs discrete token grouping, expert layers, and a SoftMax function for deterministic routing. (b) ROE Fuzzy Router (ours) integrates fuzzy logic (fuzzification and rule-based inference) over continuous features (e.g., text/image entropy, cosine similarity), enabling more nuanced and adaptive routing.

# 1 INTRODUCTION

In recent years, multimodal tasks integrating text, images, audio, and other modalities have demonstrated significant potential Fedus et al. (2022); Zhou et al. (2022). However, the heterogeneity and complexity of multimodal data pose substantial challenges for efficient model processing, including high computational costs and slow inference speeds. The Mixture of Experts (MoE) architecture effectively addresses the tension between model capacity and computational efficiency in multimodal tasks by distributing subtasks to specialized expert sub-networks Wu et al. (2025a;b); Shazeer et al.

(2017); Lepikhin et al. (2021); Dai et al. (2024). This approach significantly enhances large-scale multimodal data processing capabilities while delivering superior performance gains.

MoE models have become a pivotal research area in artificial intelligence, improving computational efficiency through multiple specialized expert modules. Nevertheless, conventional MoE designs lack skip mechanisms, requiring full computation across all layers, which limits further efficiency gains. Routing of Experts (RoE) models optimize inference speed and resource utilization by introducing layer skipping. Despite this, RoE still faces significant challenges Wu et al. (2025b).

First, its uniform skipping rate control relies on preset parameters Jelassi et al. (2025), hindering real-time adaptation to task difficulty. This results in performance fluctuations on mixed-difficulty datasets Zhou et al. (2025), particularly evident with large-scale datasets. When handling complex, diverse, and voluminous tasks—such as large-scale image recognition encompassing simple daily scenes and intricate medical images, or mixed-text classification in natural language processing involving both structured news articles and ambiguous literary expressions—static preset parameters fail to adapt flexibly to data features of varying difficulty levels. Concurrently, escalating dataset sizes drastically increase computational resource consumption Zhao et al. (2025), exacerbating performance instability. This issue has been repeatedly highlighted in recent studies, underscoring the urgency for dynamic control mechanisms.

Second, during MoE optimization, the prevalent reliance on static prior knowledge for expert selection Huang et al. (2024) exhibits critical limitations of resource utilization. Under static prior guidance, expert selection often leads to suboptimal resource allocation Shi et al. (2025): substantial computation is wasted on irrelevant experts, while task-critical experts remain under-resourced Bai et al. (2025), causing inefficient resource utilization. These challenges underscore the imperative for innovative MoE designs with enhanced adaptability and efficiency to unlock their full practical potential Jie et al. (2025).

This study proposes an enhanced RoE mechanism integrating fuzzy control with fuzzy rule interpolation (shown in Figure 1) to resolve inflexible skipping rates and inefficient prior knowledge utilization. Our method designs a sparse rule base based on prior knowledge Chao et al. (2019) and expert experience, then constructs a complete fuzzy rule set via fuzzy interpolation Lin et al. (2024b) on small datasets Song et al. (2024). The router dynamically generates routing weights through fuzzy rule interpolation, enabling path selection that is adaptive to task complexity. This approach eliminates the computational overhead of traditional weight matrices Feng & Chen (2018), significantly reducing burdens.

Experiments demonstrate that our fuzzy series methods achieve a 68.4% reduction in time and a 68.6% reduction in data during the Router stage. This directly leads to a 16.7% reduction in total training time. Meanwhile, the data volume in the Adapter and Finetune stages remains consistent with that of RoE, ensuring performance stability.

The contributions of this work are summarized as:

1. Introduces fuzzy control to build a self-augmenting dynamic regulation system, which directly infers skipping rates from the rule base and thereby enables lightweight training and inference.
2. Employs fuzzy rule interpolation to augment expert experience on small datasets, constructing a finely controllable fuzzy rule base grounded in prior knowledge.
3. Establishes a dynamically adjustable system for RoE training that autonomously refines the rule base according to task difficulty, substantially saving training time.

## 2 RELATED WORK

### 2.1 MIXTURE OF EXPERTS AND DYNAMIC ROUTING

MoE frameworks balance model capacity and computational efficiency by dynamically activating expert sub-networks. While early approaches employed static routing mechanisms, recent research emphasizes dynamic selection strategies Guo et al. (2025). For instance, Soft MoE enhances performance through the weighted fusion of multiple experts but incurs considerable computational

overhead. Conversely, Hard MoE activates only a single expert per input to reduce computation, yet this strategy may compromise performance on complex tasks Xu et al. (2025). Consequently, recent advances focus on enhancing expert specialization and optimizing dynamic routing mechanisms Liu et al. (2025).

Recent work, exemplified by Routing of Experts (RoE) Wu et al. (2025b), introduces a novel perspective by treating each transformer layer as an independent expert, enabling dynamic layer skipping through routing decisions. Key design innovations in RoE include: (1) adapter skip connections to mitigate feature discrepancies arising from skipped layers; (2) structural sparsity regularization to guide the learning of sparse execution paths; and (3) routing token alignment specifically designed for multi-turn dialogue consistency. Experimental validation demonstrates that RoE achieves a 21.3% inference acceleration on the LLaVA-1.5 Liu et al. (2023) model while maintaining stable performance.

## 2.2 FUZZY RULE INTERPOLATION

Fuzzy control theory simulates human decision-making under uncertainty. Its core lies in constructing a fuzzy rule base from prior knowledge (expert experience), using membership functions to achieve nonlinear input-output mapping. This approach has been applied in machine learning to: enhance interpretability (e.g., modeling user preferences in recommendation systems) Juang & Bui (2019); and improve decision robustness (e.g., policy optimization in reinforcement learning). To address limited expert experience, fuzzy interpolation techniques complete sparse rule bases using small datasets, validated in domains like fault diagnosis and image segmentation. The core method of fuzzy interpolation techniques involves: (1) Defining initial rules from prior knowledge (e.g., "Task difficulty increases with token count"); (2) Extending coverage to unseen input spaces via interpolation algorithms. This provides a potential solution for rule generalization in dynamic routing for Multimodal LLMs (MLLMs) Wang et al. (2025), while existing RoE relies on data-driven $W_r$ weight learning, fuzzy interpolation can generate adaptive control policies from minimal prior rules, potentially reducing training costs.

## 3 METHOD

### 3.1 OVERVIEW OF FUZZY ROUTER

We propose a Fuzzy Router, shown in Figure 2, to dynamically decide whether to execute the full transformer layer or a lightweight adapter for each token, balancing accuracy and efficiency. The method consists of four core modules: **(1) Fuzzy Antecedent Computation:** Extract fuzzy attributes (uncertainty, dispersion, cross-modal relevance) from multimodal features. **(2) Rule-Based Inference:** Map fuzzy attributes to routing decisions using a learnable rule base, with interpolation inference for no matching rule cases. **(3) Adaptive Rule Refinement:** The Fuzzy Router's adaptive rule base refinement uses two-stage training: pretraining (freezing backbone, updating rule consequents to minimize task loss) and end-to-end fine-tuning (unfreezing all parameters, with total loss combining task loss, fuzzy skip loss, and sparsity loss) to co-optimize the rule base and model parameters. **(4) Sparse region rule enhancement:** For sparse regions with low rule coverage and high error (>20%), a rule augmentation strategy is proposed: identify such regions by interpolation error, then generate new rules (assigning linguistic categories and setting consequents as the mode of optimal skip rates) to fill gaps and reduce the error rates of fuzzy interpolation inference results.

### 3.2 FUZZY ANTECEDENT COMPUTATION

The fuzzy antecedent translates high-dimensional multimodal features into interpretable low-dimensional fuzzy attributes, capturing both single-modal uncertainty and cross-modal semantic relationships. Specifically, we define three antecedent variables $(x_1, x_2, x_3)$, corresponding to text uncertainty, image uncertainty, and cross-modal relevance.

**Text Feature Fuzzification** $(x_1)$: We first quantify the fuzzy properties of text features, such as uncertainty and dispersion, by defining $x_1$. For entropy calculation, normalize text features $x_{\text{Text}} \in$

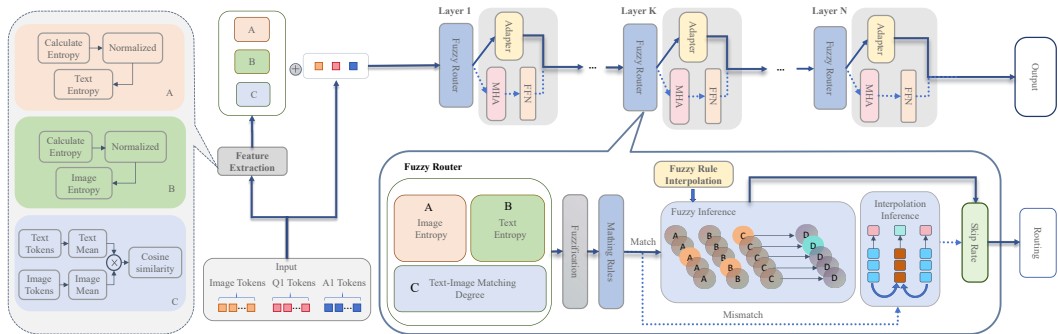

Figure 2: Schematic diagram of the fuzzy ROE principle. Input tokens first derive three core features: (A) Normalized Text Entropy, (B) Normalized Image Entropy, and (C) Text-Image Matching Degree (via cosine similarity of token mean representations). Each layer embeds a Fuzzy Router, which processes these features through fuzzification, rule matching, and fuzzy inference with fuzzy rule interpolation inference as a fallback for unmatched cases. The inference output, combined with a learnable Skip Rate, decides token routing. Routed tokens then flow through the adapter-based skip connection or the default Transformer layer, enabling dynamic, uncertainty-aware computation for multi-modal inputs.

$\mathbb{R}^{B \times T_{\text{Text}} \times D}$ along the dimension axis using softmax:

$$\text{attn\_sim}(d) = \frac{\exp(x_{\text{Text}}(d))}{\sum_{k=1}^{D} \exp(x_{\text{Text}}(k))}. \tag{1}$$

Compute information entropy to measure uncertainty:

$$\text{entropy}(x_{\text{Text}}) = -\sum_{d=1}^{D} \text{attn\_sim}(d) \cdot \log(\text{attn\_sim}(d) + 10^{-8}). \tag{2}$$

Normalize to $[0, 1]$ using batch-wise maximum entropy:

$$x_1 = \frac{\text{entropy}(x_{\text{Text}})}{\max(\text{entropy}(x_{\text{Text}})) + 10^{-6}}. \tag{3}$$

**Image Feature Fuzzification** ($x_2$): Analogously, we define $x_2$ to capture the uncertainty of image features. The same entropy-based procedure is applied to the image tensor $x_{\text{Image}} \in \mathbb{R}^{B \times T_{\text{Image}} \times D}$, producing a normalized fuzzy attribute parallel to $x_1$.

**Cross-Modal Relevance** ($x_3$): Finally, cross-modal relevance $x_3$ evaluates the semantic alignment between the image and text modalities. We first compress features along the token axis:

$$\begin{aligned} \bar{x}_{\text{image}} &= \text{Mean}(x_{\text{image}}, \dim = 1) \in \mathbb{R}^{B \times D}, \\ \bar{x}_{\text{text}} &= \text{Mean}(x_{\text{text}}, \dim = 1) \in \mathbb{R}^{B \times D}. \end{aligned} \tag{4}$$

Subsequently, we compute the cosine similarity between the modality-specific representations:

$$x_3 = \cos(\bar{x}_{\text{image}}, \bar{x}_{\text{text}}), \quad x_3 \in [-1, 1], \tag{5}$$

where $\bar{x}_{\text{image}}$ and $\bar{x}_{\text{text}}$ denote the aggregated image and text embeddings, respectively. The similarity score $x_3$ is then broadcast to align with the dimensionality of $x_1$ and $x_2$ ($B \times T_{\text{txt}}$). Together, the triplet of antecedent variables ($x_1, x_2, x_3$) provides the input to the fuzzy rule system, forming the foundation for subsequent dynamic routing decisions.

## 3.3 Fuzzy Routing Inference Mechanism

The fuzzy routing mechanism converts crisp antecedent values into fuzzy attributes (via fuzzification) and maps them to routing decisions using a rule base, with interpolation for sparse regions.

**Fuzzification of Antecedent Variables**: Each antecedent $(x_1, x_2, x_3)$ is fuzzified into linguistic categories using triangular membership functions. Fuzzification produces membership degrees $\mu_c(x_i)$ (where $c$ denotes a linguistic category), bridging continuous feature values and discrete rule conditions.

**Fuzzy Rule Matching**: A predefined rule library maps fuzzified antecedent categories to a routing factor $f$ (range: $[0, 1]$, where a higher $f$ indicates a stronger necessity to retain the original layer). Rules follow the form: If $x_1$ is [Category1], $x_2$ is [Category2], and $x_3$ is [Category3], then $f = c$, where [Category1–3] are linguistic terms (e.g., "low," "medium") and $c$ is the consequent fuzzy factor. For each token, we first check if its fuzzified antecedent categories match any rule in the library. If a match is found, the corresponding $f$ is directly adopted.

### 3.4 INTERPOLATION INFERENCE FOR UNMATCHED CASES

When the input antecedents do not match any predefined rules, we adopt fuzzy rule interpolation (FRI) based on the scale-and-move transformation approach (commonly known as T-FRI) Huang & Shen (2008); Chen et al. (2020); Li et al. (2021). To enable this process, the antecedents and consequents are first formalized as fuzzy sets; the closest rules are then identified in the rule base; and finally, an intermediate rule is constructed through interpolation combined with scale-and-move transformation.

**Fuzzy Set Representation**: To support interpolation, rules' antecedents and consequents are expressed as polygonal fuzzy sets (e.g., triangular or trapezoidal). For a fuzzy set $A = (a_0, \cdots, a_{n-1})$ with $n$ key odd points, its representative value $Rep(A)$ (capturing overall location) is calculated as:

$$Rep(A) = \sum_{i=0}^{n-1} w_i a_i, \tag{6}$$

where $w_i$ are weights. For average $Rep$, all $w_i$ are equal; for weighted $Rep$, weights increase with membership values.

**Selecting Closest Rules**: Given this representation, the next step is to identify existing rules most similar to the unmatched input. For input $A^*$, distance to a rule antecedent $A_i$ is defined as: $d(A_i, A^*) = |Rep(A_i) - Rep(A^*)|Y$, which is normalized by the attribute domain range. We then select $n \geq 2$ rules with the smallest normalized distances.

**Constructing Intermediate Rule**: Based on the selected rules, an intermediate rule is constructed to approximate the unmatched case. **(1) Weights Calculation**: For selected rules $k$ and attribute $i$, normalized weights are defined as:

$$w'_{ki} = \frac{1/d(A_{ki}, A_k^*)}{\sum_j (1/d(A_{ji}, A_j^*))}. \tag{7}$$

**(2) Intermediate Antecedent**: Aggregate selected antecedents to form $A'' = \sum_k w'_{ki} A_{ki}$, then shift to $A'$ such that $Rep(A') = Rep(A^*)$. **(3) Intermediate Consequent**: Using average antecedent weights $w'_{ai}$, construct $B'' = \sum_k w'_{ai} B_k$, then shift to $B'$.

**Scale and Move Transformations**: Finally, the intermediate rule is refined through scale and move operations. **(1) Scale Transformation**: Adjust support lengths of $A'$ to match $A^*$ via scale rates:

$$s_i = \frac{a^*_{n-1-i} - a^*_i}{a'_{n-1-i} - a'_i}. \tag{8}$$

**(2) Move Transformation**: Shift scaled $A'$ to $A^*$ using move ratios $M_i$, preserving support lengths and $Rep$. **(3) Apply to Consequent**: Apply the same transformations to $B'$ to generate interpolated consequent $B^*$, which retains convexity and normality.

If a crisp value is needed, defuzzify $B^*$ (e.g., via centroid method). The resulting fuzzy factor is then converted to routing weights using a temperature parameter to control distribution sharpness.

### 3.5 ADAPTIVE RULE BASE REFINEMENT IN TRAINING

While the initial rule base, built on prior expert knowledge, has shown reasonable performance, it still lacks sufficient accuracy to meet the demands of complex tasks. Thus, further refinement

of these rules through training is essential to enhance their precision and alignment with complex inference needs. The Fuzzy Router's training process integrates rule-based evolution and model parameter optimization, enabling dynamic adaptation to sparse feature regions and task complexity.

**Two-Stage Training with Rule Evolution**: Training is decomposed into two stages to stabilize convergence:

**(1) Pretraining (Rule Initialization)**: Backbone parameters of the Multimodal Large Language Model (MLLM) are frozen to mitigate overfitting. We minimize the task loss ($L_{\text{task}}$) by updating only the consequent fuzzy factors (f-values) in the fuzzy rule base; gradients are backpropagated through the fuzzy inference pipeline to align initial routing decisions with ground-truth task outputs. This establishes a preliminary task-compliant routing strategy for the rule base, laying the groundwork for subsequent joint optimization of feature learning and routing logic.

**(2) End-to-End Fine-Tuning**: Both MLLM backbone and fuzzy rule base parameters are unfrozen for joint optimization, integrating the backbone's task-specific feature learning and the rule base's routing logic refinement.

The **Sparse Loss for Fuzzy Router** ($L_{\text{sparse,FuzzyRouter}}$) is adapted from the sparse loss ($L_{\text{sparse}}$) in Routing of Experts (RoE) and tailored to fuzzy logic-based routing, with the form:

$$L_{\text{sparse, FuzzyRouter}} = \mathbb{E}\left[\max\left(t - \frac{1}{T}\sum_{i=1}^{T}p_i, 0\right)\right], \tag{9}$$

where $\mathbb{E}$ denotes batch expectation, $t$ is the target average skip rate, $T$ is the number of tokens per sample; this loss constrains the average skip rate to approximate $t$, promoting efficient adapter usage and reducing redundant computations.

The **Total Loss ($L_{\text{total}}$)** integrates task-specific loss and the sparse loss as:

$$L_{\text{total}} = L_{\text{task}} + \alpha L_{\text{sparse, FuzzyRouter}}, \tag{10}$$

where $L_{\text{task}}$ drives the backbone to learn discriminative multimodal features (text entropy, image entropy, cross-modal similarity)—key inputs for the rule base's routing inference; $\alpha$ (0.1–0.5, validated experimentally) balances loss weights, ensuring $L_{\text{sparse,FuzzyRouter}}$ reduces redundancy without compromising task performance.

A feedback loop forms: $L_{\text{task}}$-optimized backbone features enable rational rule-based routing; $L_{\text{sparse,FuzzyRouter}}$-refined rule base controls average skip rates to reduce computational overhead and aid backbone feature learning. This synergy achieves the core goal: enhancing multimodal learning efficiency while preserving accuracy.

## 3.6 SPARSE REGION RULE AUGMENTATION

For feature regions with low rule coverage (sparse regions) and high inference error rates ($>20\%$), we introduce a rule augmentation strategy, shown in Figure 3, to fill coverage gaps: In these sparse regions, where existing rules are scarce, fuzzy rule interpolation inference—used as a fallback when no matching rules exist—performs poorly with low accuracy. Such unreliable interpolated results make direct rule supplementation necessary, reducing over-reliance on interpolation and enhancing inference robustness in these under-covered areas.

**Identifying High-Error Sparse Regions**: We partition the 3D antecedent space ($x_1$: text entropy, $x_2$: visual entropy, $x_3$: cross-modal similarity) into a grid and compute interpolation errors, thus, the "error metric" is defined as:

$$\epsilon(x) = |f_{\text{interp}}(x) - f_{\text{opt}}(x)|, \tag{11}$$

where $f_{\text{opt}}(x)$ is the optimal skip rate determined via validation set distillation. For each sample $x$, we compare both paths and define:

$$f_{\text{opt}}(x) = \begin{cases} 1, & \text{if } \mathcal{L}_{\text{adapter}} \leq \mathcal{L}_{\text{full}} + \tau_{\text{diff}}(x) \text{ and } \text{sim}(H_{\text{full}}, H_{\text{adapter}}) \geq 0.95 \\ 0, & \text{otherwise} \end{cases} \tag{12}$$

where $\tau_{\text{diff}}(x)$ is a difficulty-aware tolerance calibrated on the validation set: higher for simple samples (e.g., 0.08) and lower for complex ones (e.g., 0.02), enabling aggressive skipping when accuracy impact is negligible relative to speedup.

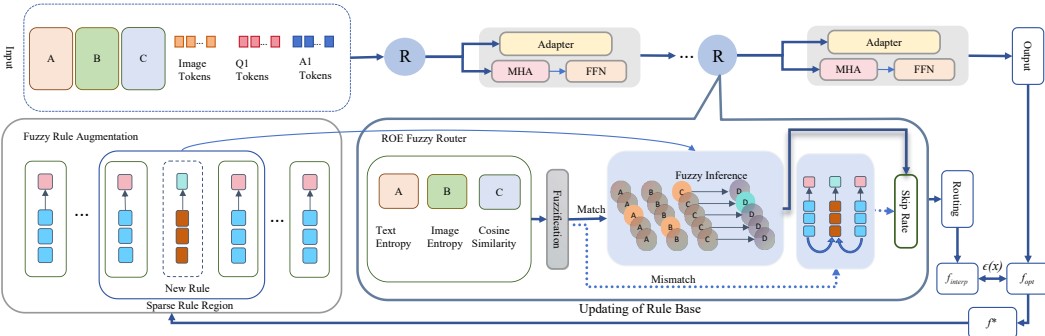

Figure 3: Fuzzy rule base expansion for sparse regions. Within the rule space, sparse regions are identified. For such regions, new rules are generated: antecedents adopt linguistic labels via membership-degree maximization at the region's center, and consequents are set as the mode of optimal skip rates within the region. This process fills coverage gaps and reduces errors from fuzzy rule interpolation inference.

In addition, a region is marked as "high-error sparse", which is defined as: if $\epsilon(x) > 0.2$ (local error exceeds 20%), and the average error in its $3 \times 3 \times 3$ neighborhood also exceeds 0.2.

**Generating New Rules for Sparse Regions**: For each high-error region, we generate a new rule to fill the coverage gap.

**(1) Antecedent Fuzzy Set Assignment:** For the region's center $(x_1^*, x_2^*, x_3^*)$, assign linguistic categories via membership degree maximization. **(2) Consequent Skip Rate Calculation:** The rule's consequent $f^*$ is the statistical mode of optimal skip rates in the region:

$$f^* = \mathrm{mode}\left\{|f_{\mathrm{opt}}(x)| \mid x \in R_{\mathrm{high\text{-}error}}\right\}, \tag{13}$$

where $R_{\mathrm{high\text{-}error}}$ is the set of samples in the high-error region. The fuzzy set for $f^*$ uses a triangular membership function with its vertex at $f^*$ to ensure smooth transitions with adjacent rules.

This two-stage training and sparse region augmentation ensure the Fuzzy Router's rule base evolves from coarse to fine, balancing coverage, efficiency, and task performance. The optimization with fuzzy rule interpolation, shown in Figure 3, continuously refines both the model's feature representations and its decision logic.

## 4 EXPERIMENTATION

To comprehensively validate the effectiveness of the proposed fuzzy routing mechanism, we design a series of experiments across multiple multimodal large language models. This section first introduces the datasets and evaluation metrics employed, followed by the implementation details and experimental results. Together, these analyses provide a systematic assessment of both efficiency and accuracy improvements achieved by our method.

### 4.1 DATASETS AND METRICS

The benchmarks used in this paper are five common vision-language and five recently proposed MLLM benchmarks. The common VL benchmarks include VQAv2 Goyal et al. (2017), GQA Hudson & Manning (2019), ScienceQA Lu et al. (2022), VizWiz Bigham et al. (2010) and TextVQA Singh et al. (2019).

### 4.2 IMPLEMENTATION DETAILS

We apply our Fuzzy Router to the same three popular MLLMs as RoE, namely LLaVA-1.5 Liu et al. (2024), LLaVA-HR Luo et al. (2024) and VILA Lin et al. (2024a), and term the new models as Fuzzy-LLaVA, Fuzzy-LLaVA-HR and Fuzzy-VILA, respectively.

For consistency with RoE, the hidden dimension of adapter connections in our Fuzzy Router framework is kept at 1,024. We introduce a hyper-parameter $\beta$ (set to 0.3) to control the sensitivity of the fuzzy logic-based decision mechanism, balancing between skip accuracy and computational efficiency. For three-stage training, the data sampling strategy for the Adapter and Finetune stages remains consistent with RoE.

Specifically, the Adapter and Finetune stages use the same proportions of the 665k instruction data from LLaVA-1.5 as their RoE counterparts. Notably, for the Router stage, we employ interpolation-based training instead of full sample training, thus reducing the data usage to 32.8%, 31.3% and 34.3% of the 67k Router data for the three RoE-models.

During training, the MLLMs are optimized with a learning rate of $2 \times 10^{-6}$, consistent with RoE. The adapters share the same learning rate of $4 \times 10^{-4}$ as in RoE, while the fuzzy routers are updated with a slightly lower learning rate of $3 \times 10^{-4}$ to stabilize training with reduced data. The training epoch is set to 1 with early stopping applied. All other settings are kept identical to the original MLLMs and RoE. More implementation details are available in our code project.

## 4.3 EXPERIMENTAL RESULTS

We conduct extensive experiments on four benchmark datasets (**ScienceQA**, **GQA**, **MMB**, and **SEED**) to evaluate the effectiveness of our Fuzzy Router.

Table 1 compares Fuzzy methods with baselines and RoE methods on three MLLMs, evaluating accuracy (Acc.), speed (samples/sec), and actual skip rate (Skip) across datasets to verify the effectiveness of the Fuzzy strategy. Others' data includes baselines with 0% skip rate, slower speed, and baseline accuracy, as well as RoE methods with fixed target skip rates (10%, 20%), moderate speed improvement, and unstable accuracy. Our data (Fuzzy methods) features adaptive skip rates varying with task difficulty (e.g., Fuzzy-LLaVA: 22.08% on ScienceQA vs 9.39% on SEED), the fastest speed among all methods, and stable accuracy—with the highest average in LLaVA (63.3). Our advantages lie in flexible skip rates adapting to task difficulty, optimal speed, and stable accuracy, verifying that the Fuzzy strategy can flexibly adjust skip rates by task difficulty, boosting speed while maintaining high accuracy.

Table 1: Results of Fuzzy methods with different skip rates on three MLLMs. "Acc.", "Speed" and "Skip" denote accuracy, samples per second, and the actual skip rate, respectively.

| Method | ScienceQA | | | GQA | | | MMB | | | SEED | | | Average | | |
|---|---|---|---|---|---|---|---|---|---|---|---|---|---|---|---|
| | Acc. | Speed | Skip | Acc. | Speed | Skip | Acc. | Speed | Skip | Acc. | Speed | Skip | Acc. | Speed | Skip |
| LLaVA (baseline) | 66.8 | 7.55 | 0.00% | **62.0** | 6.99 | 0.00% | 64.3 | 8.37 | 0.00% | 58.6 | 8.33 | 0.00% | 62.9 | 7.81 | 0.00% |
| RoE-LLaVA 10% | 68.4 | 7.65 | 10.26% | 61.4 | 7.07 | 4.59% | 64.3 | 9.62 | 20.55% | 58.2 | 8.41 | 9.04% | 63.1 | 8.19 | 7.77% |
| RoE-LLaVA 20% | **68.7** | 9.15 | 20.55% | 61.3 | 7.52 | 7.86% | 64.6 | 9.88 | 23.64% | 57.8 | 9.85 | 24.52% | 63.1 | 9.10 | 19.15% |
| Fuzzy-LLaVA | 68.4 | **10.26** | 22.08% | 61.9 | **8.70** | 12.32% | **64.8** | **11.08** | 19.81% | **59.0** | **10.16** | 9.39% | **63.5** | **10.20** | 15.90% |
| VILA (baseline) | 68.2 | 8.27 | 0.00% | 62.3 | 8.03 | 0.00% | **68.9** | 8.51 | 0.00% | 58.36 | 8.36 | 0.00% | 64.4 | 8.29 | 0.00% |
| RoE-VILA 10% | **69.5** | 8.39 | 9.19% | 62.2 | 8.01 | 4.83% | 67.6 | 8.63 | 10.59% | **61.3** | 8.50 | 11.41% | **65.2** | 8.38 | 11.94% |
| RoE-VILA 20% | 68.4 | 10.49 | 23.93% | 61.1 | 8.20 | 12.02% | 67.8 | 10.37 | 19.57% | 61.2 | 9.85 | 22.34% | 64.6 | 9.73 | 19.45% |
| Fuzzy-VILA | 69.2 | **11.58** | 21.81% | **62.4** | **9.34** | 11.52% | 67.9 | **11.48** | 20.20% | 60.6 | **10.90** | 8.87% | 65.0 | **10.80** | 15.60% |
| LLaVA-HR (baseline) | 65.1 | 4.82 | 0.00% | **64.2** | 4.87 | 0.00% | 64.9 | 4.76 | 0.00% | **64.2** | 3.74 | 0.00% | 64.6 | 4.55 | 0.00% |
| RoE-LLaVA-HR 10% | 67.4 | 4.96 | 7.96% | 62.5 | 5.01 | 7.65% | 64.6 | 4.82 | 6.96% | 62.2 | 3.86 | 8.43% | 64.2 | 4.66 | 7.68% |
| RoE-LLaVA-HR 20% | 56.1 | 4.97 | 12.77% | 60.8 | 5.09 | 11.07% | 52.9 | 4.89 | 10.63% | 58.8 | 3.92 | 13.62% | 57.2 | 4.72 | 12.02% |
| Fuzzy-LLaVA-HR | **67.5** | **6.10** | 19.50% | 63.2 | **6.23** | 9.80% | **65.5** | **6.07** | 18.60% | 62.4 | **4.97** | 7.50% | **64.7** | **5.81** | 13.90% |

Table 2 shows average actual skip rates (%) across datasets, highlighting skip rate variation with task difficulty. Others' data (RoE-LLaVA 20%) has skip rates unrelated to difficulty (e.g., harder SEED has a higher skip rate than easier ScienceQA). Our Fuzzy methods show rational variation, higher skip rates for easier tasks (ScienceQA: 21.8%-22.1%) and lower for harder ones (SEED: 7.5%-9.4%), matching difficulty. Advantage lies in adaptive skip rates with task difficulty, verifying the Fuzzy strategy's ability to adjust skip rates by task difficulty.

Table 3 demonstrates that the Router stage of Fuzzy models uses only 32.8%, 31.3% and 34.3% of the 67k Router data, reducing training time for the router by 68.4% (e.g., from 18.7 to 5.9 for LLaVA). This directly shortens the total training time 16.7%(Fuzzy-LLaVA takes 80.2 time units vs. 93.6 for RoE-LLaVA). The reduced computational overhead during inference (reflected by "Speed" in Table 1) translates to faster token processing. For example, Fuzzy-LLaVA achieves a speed of 10.20 (vs. 9.10 for RoE-LLaVA 20%) with a lower average skip rate, indicating more efficient

Table 2: Average actual skip rates (%) on different datasets. The fuzzy router adapts to dataset difficulty, skipping more for easier tasks (e.g., ScienceQA) and less for harder tasks (e.g., SEED).

| Dataset | RoE-LLaVA 20% | Fuzzy-LLaVA | Fuzzy-VILA | Fuzzy-LLaVA-HR |
|---|---|---|---|---|
| ScienceQA | 20.55% | 22.1% | 21.8% | 19.5% |
| GQA | 7.86% | 12.3% | 11.5% | 9.8% |
| MMB | 23.64% | 19.8% | 20.2% | 18.6% |
| SEED | 24.52% | 9.4% | 8.9% | 7.5% |

token throughput per unit time. Despite the efficiency gains, our Fuzzy models maintain or slightly improve accuracy compared to RoE. For instance, Fuzzy-LLaVA achieves an average accuracy of 63.5, outperforming RoE-LLaVA 20% (63.1) . This confirms that the interpolation-based router does not compromise performance while reducing costs.

Table 3: The training costs of Fuzzy methods on three MLLMs. "Adapter" and "Router" denote the adapter warmup and router warmup of RoE or Fuzzy Router. We use A800 GPU hours to measure the training time (Time). "Data" denotes the number of training examples.

| Method | Adapter | | Router | | Finetune | | Total | |
|---|---|---|---|---|---|---|---|---|
| | Time | Data | Time | Data | Time | Data | Time | Data |
| VILA | 0.0 | 0 | 0.0 | 0 | 120.9 | 1M | 120.9 | 1M |
| RoE-VILA | 25.3 | 100k | 18.7 | 67k | 49.6 | 166k | 93.6 | 333k |
| Fuzzy-VILA | 24.8 | 100k | 6.2 | 22k | 50.1 | 166k | 81.1 | 288k |
| LLaVA | 0.0 | 0 | 0.0 | 0 | 82.6 | 665k | 82.6 | 665k |
| RoE-LLaVA | 25.3 | 100k | 18.7 | 67k | 49.6 | 166k | 93.6 | 333k |
| Fuzzy-LLaVA | 25.1 | 100k | 5.9 | 21k | 49.2 | 166k | 80.2 | 287k |
| LLaVA-HR | 0.0 | 0 | 0.0 | 0 | 128.4 | 665k | 128.4 | 665k |
| RoE-LLaVA-HR | 39.1 | 100k | 29.0 | 67k | 76.7 | 166k | 144.8 | 333k |
| Fuzzy-LLaVA-HR | 38.7 | 100k | 8.3 | 23k | 77.2 | 166k | 124.2 | 289k |

Table 4 further validates the effectiveness of our interpolation-based implementation. Compared with the "Fuzzy (only)" variant, our method enriches the rule base through interpolation; despite a slight increase in time overhead, it achieves more precise skip rate control (reflected by adaptive skip rates across datasets). When compared to the original LLaVA baseline, our interpolated Fuzzy Router maintains comparable accuracy (63.5 vs. 62.9) while boosting speed by 30.6% (10.20 vs. 7.81). These results demonstrate that the interpolation mechanism enables a better balance between accuracy and efficiency—trading marginal time cost for substantial precision gains and adaptive control over skipping behavior.Removing any antecedent weakens routing quality. The average accuracy drops to 62.9% (w/o $x_1$), 63.3% (w/o $x_2$), and 61.7% (w/o $x_3$), showing that all three signals contribute meaningfully. The largest decline appears when discarding cross-modal similarity $x_3$, which reduces GQA/SEED performance and produces imbalanced skip behavior (easy tasks skip more, hard tasks skip less). This indicates that unimodal entropy alone is insufficient, and cross-modal consistency is crucial for distinguishing genuinely difficult samples. Removing $x_1$ or $x_2$ causes smaller but stable drops, suggesting both modality-specific uncertainties remain necessary for reliable fine-grained routing.

Table 4: Ablation studies on key components of Fuzzy Router. "Fuzzy (only)" uses fuzzy control without rule interpolation; "Reg20%" incorporates 20% strength sparse regularization.Rows marked "w/o x_i" remove the corresponding antecedent feature from the Fuzzy-Router (x_1=text entropy, x_2=image entropy, x_3=cross-modal similarity).

| Method | ScienceQA | | | GQA | | | MMB | | | SEED | | | Average | | |
|---|---|---|---|---|---|---|---|---|---|---|---|---|---|---|---|
| | Acc. | Speed | Skip | Acc. | Speed | Skip | Acc. | Speed | Skip | Acc. | Speed | Skip | Acc. | Speed | Skip |
| LLaVA | 66.8 | 7.55 | 0.00% | 62.0 | 6.99 | 0.00% | 64.3 | 8.37 | 0.00% | 58.6 | 8.33 | 0.00% | 62.9 | 7.81 | 0.00% |
| + Fuzzy (only) | 60.1 | 10.02 | 5.20% | 55.3 | 9.25 | 4.80% | 58.0 | 10.85 | 8.60% | 51.7 | 10.70 | 7.50% | 56.1 | 10.21 | 6.53% |
| + Reg20% | 64.3 | 8.63 | 15.48% | 59.6 | 7.57 | 7.00% | 63.8 | 9.32 | 17.89% | 56.6 | 9.18 | 18.49% | 61.1 | 8.59 | 14.72% |
| + Fuzzy-Router | 68.4 | 10.26 | 22.08% | 61.9 | 8.70 | 12.32% | 64.8 | 11.08 | 19.81% | 59.0 | 10.16 | 9.39% | 63.5 | 10.20 | 15.90% |
| + Fuzzy-Router (w/o $x_1$) | 67.6 | 10.14 | 21.12% | 61.4 | 8.65 | 11.80% | 64.3 | 11.05 | 18.90% | 58.4 | 10.08 | 8.90% | 62.9 | 9.98 | 15.18% |
| + Fuzzy-Router (w/o $x_2$) | 68.0 | 10.18 | 21.60% | 61.6 | 8.68 | 11.90% | 64.7 | 11.04 | 19.60% | 58.8 | 10.12 | 9.00% | 63.3 | 10.00 | 15.53% |
| + Fuzzy-Router (w/o $x_3$) | 66.2 | 9.90 | 24.50% | 60.1 | 8.50 | 14.00% | 63.4 | 10.80 | 17.10% | 57.2 | 9.95 | 7.00% | 61.7 | 9.54 | 15.65% |

## 5 CONCLUSION

In this work, we introduced a **Fuzzy Router** that integrates fuzzy rule interpolation (FRI) into Mixture of Experts (MoE) and Routing of Experts (RoE) frameworks to enhance multimodal learning efficiency. Our approach addresses two fundamental limitations of existing designs: the inability to adapt skip rates to task complexity and the inefficiency of static expert selection. By combining dynamic skip-rate regulation, sparse FRI for comprehensive rule coverage, and adaptive refinement through two-stage training, the proposed method achieves substantial improvements in both computational efficiency and performance stability. Extensive experiments demonstrate that our FRI-based router reduces routing time and data usage by over 68%, cutting overall training costs by 16.7%, while maintaining or even improving model accuracy. These results confirm that fuzzy logic provides an effective mechanism for adaptive resource allocation, striking a favorable balance between computation and accuracy in heterogeneous multimodal tasks.

For future work, we will explore richer membership functions for stronger cross-modal modeling, extend the FRI-enhanced routing paradigm to non-transformer MoE architectures, and investigate continual rule adaptation under distribution shifts. These directions aim to further improve the adaptability and scalability of multimodal MoE/RoE systems.

## ETHICS STATEMENT

This research, which optimizes multimodal learning efficiency via a Fuzzy Router based on fuzzy rule interpolation, uses publicly available benchmark datasets (e.g., VQAv2, GQA, ScienceQA) with clear usage licenses and no sensitive/private information, ensuring compliance with dataset ethical guidelines; we balanced data distribution across task difficulties to reduce model bias and explicitly prohibit the Fuzzy Router's use in harmful scenarios (e.g., misleading information generation, unauthorized privacy analysis), adhering to academic ethics and responsible AI principles.

## REPRODUCIBILITY STATEMENT

To ensure reproducibility of our research on the Fuzzy Router, we detailed the Fuzzy Router's core modules (antecedent computation, rule interpolation, adaptive refinement), experimental settings (base models like LLaVA-1.5, hyperparameters such as learning rates, "Nvidia A800" GPU-based training time measurement), and will publicly release source code, pretrained model weights, data preprocessing tools, and dependency configurations post-publication, along with a step-by-step guide for environment setup, training, and evaluation.

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
