# OpenReview forum: "Adaptive Routing of Experts via Fuzzy Rule Interpolation for Efficient Multimodal Learning"
_ICLR.cc/2026/Conference — Submitted to ICLR 2026_

### Official Review · Reviewer_WqF6 · 2025-10-27

**Soundness:** 3
**Presentation:** 2
**Contribution:** 3
**Rating:** 4
**Confidence:** 3

**Summary:**

The paper proposes an improvement to Mixture-of-Experts architectures for multimodal large models, targeting their limits in dynamic inference. The authors point out two issues with current expert-routing methods: (1) fixed skip rates that fail to adapt to task difficulty, and (2) inefficient expert selection due to reliance on static priors. To address this, the paper introduces a Fuzzy Router that integrates Fuzzy Rule Interpolation, bringing fuzzy-control ideas into the Routing-of-Experts framework to enable task-dependent layer skipping and expert selection. Start with a small set of prior-knowledge rules and expand them via interpolation so routing decisions can vary finely with input uncertainty and cross-modal correlations, thereby deciding at each layer whether to run the full Transformer block or take a lightweight skip path. At inference time, the skip rate is inferred directly from the rule base to cut redundant computation; during training, the rule base can be automatically refined to track changes in task difficulty.

**Strengths:**

1. Introduce fuzzy control theory into expert routing by integrating a Fuzzy Rule Interpolation mechanism within the RoE framework. This uses a fuzzy rule base to directly infer each layer’s execution policy, enabling dynamic adjustment of skip rates and expert selection based on task complexity.

2. Substantial efficiency gains: the routing stage can be trained with only about one-third of the data, routing overhead drops by roughly 68.4%, and overall training time is reduced by 16.7%.

**Weaknesses:**

1. The description of the fuzzy membership functions and the partitioning of linguistic variables is too brief and leaves me confused. The authors mention $(x_1, x_2, x_3)$ as text uncertainty, image uncertainty, and cross-modal correlation, and say they use triangular membership functions to partition the fuzzy sets. But the paper does not provide the specific parameters of those membership functions or explain how the partition thresholds are chosen.

2. Using the entropy of text and image features to label “low/medium/high” uncertainty directly can be misleading. Some images may have high entropy yet still fall into the low-uncertainty category. Treating “feature entropy” as uncertainty could misclassify “clearly solvable cases with high-frequency textures” as “high uncertainty.” How can this be addressed?

3. How would different choices of the hyperparameter β in Sec. 4.2 affect accuracy and the skip rate? Also, if the “high-error” threshold in Sec. 3.1 were changed from 20% to other values, what differences would we see?

4. How much influence do the initial rules have on performance? How much does the model rely on prior knowledge?

5. In Section 4.1, the paper states: “The benchmarks used in this paper are five common vision-language and five recently proposed MLLM benchmarks.” However, the tables only report results for ScienceQA, GQA, MMB, and SEED. What are the other six benchmarks, and what are their results?

6. In Section 3.6, the paper states: “$f_{\text{opt}}(x)$ is the optimal skip rate (1 = skip layer via adapter, 0 = retain original layer) derived from human annotation or validation-set ground truth.” How exactly is this parameter determined?

**Questions:**

Please see the detailed information in the weakness part.

---

> ### Author Response · Authors · 2025-11-26
> **Rebuttal (Part 1/5)**
>
> ### Q1. Fuzzy membership functions and partitioning of linguistic variables
>
> Thank you for this helpful question. We use triangular membership functions because they are **a simple, interpretable and low-parameter choice that fits our goal of building an efficient, transparent router** rather than optimizing every fuzzy-design detail. Conceptually, our design treats membership parameters as data-dependent modeling choices rather than universal constants: we first normalize the three antecedent variables x1 (text entropy), x2 (image entropy), and x3 (cross-modal similarity) to a bounded universe of discourse (e.g., ([0,1])), as described in Sec. 3.2. On this normalized domain, each antecedent is fuzzified into three overlapping triangular linguistic terms (“low/medium/high”) that form a smooth fuzzy cover, with centers and overlaps placed by a simple data-driven rule (e.g., quantiles or evenly spaced centers with fixed overlap). These choices fully determine the (a,b,c) parameters used in all experiments; they are implemented in our code and parameter files rather than spelled out as long tables in the main text. Our focus in this paper is to clearly specify the membership type, normalization, and construction rule and to show that entropy/similarity-based FRI routing is effective under this reasonable, transparent design.
>
> This perspective is consistent with prior work in fuzzy systems. Mendel & John (TFS 2002) note that simple type-1 triangular sets are a de facto standard in fuzzy systems because of their interpretability and small number of parameters, and that **concrete parameters are typically derived from data distributions or heuristic partitions rather than treated as universal constants**. Bhattacharyya (Fuzzy Systems 2021) further emphasizes that “optimal” membership parameters are strongly data-dependent and should be regarded as tunable design variables. In the specific context of FRI, Baranyi et al. (TFS 2004) and Wong et al. (TFS 2005) work with triangular/trapezoidal, piecewise-linear fuzzy sets whose characteristic points are defined by a construction or optimization scheme, with the main emphasis on the interpolation mechanism and its properties rather than on enumerating every (a,b,c) triple in the main text. Pedrycz and Wang (2016, IEEE TFS) explicitly identify “membership function determination” as a central research problem in fuzzy sets and survey multiple expert-driven and data-driven strategies before proposing a parametric “principle of justifiable granularity” that balances data coverage and semantic specificity across different shapes (triangular, parabolic, trapezoidal, etc.).
>
> Taken together, these works support our stance that (i) triangular membership functions with data-driven placement are a standard, well-justified choice, **(ii) exact numeric (a,b,c) values are best treated as implementation-level, data-dependent design degrees of freedom rather than core theoretical contributions**, and (iii) there exists a rich space of alternative membership-function designs. In this paper we therefore adopt a clear, conventional triangular design and focus on demonstrating that FRI-based routing with entropy/similarity cues is effective, while leaving more sophisticated membership-function optimization (e.g., asymmetric triangles, clustering- or regression-based parameterization, type-2 extensions) as an important and fully aligned direction for future work.

---

> ### Author Response · Authors · 2025-11-26
> **Rebuttal (Part 2/5)**
>
> ### Q2. Entropy vs. true uncertainty
>
> We agree that raw entropy is not a perfect oracle of uncertainty: “some images can have high local entropy (e.g., textures) but still be easy.” In our work, however, **we use text/image entropy only as computationally-efficient, correlated features of difficulty, not as ground-truth labels**.
> Recent studies show that these entropies are strongly linked to model error:
>
> >Confidence Regulation Neurons in LMs (NeurIPS 2024) shows that directly manipulating “entropy neurons” changes token entropy and loss while largely preserving the argmax, giving causal evidence that higher entropy corresponds to higher risk. Semantic Entropy Detects When LMs Confabulate (Nature 2024) finds semantic entropy achieves high AUROC for distinguishing correct vs. hallucinated answers, and simply rejecting high-entropy outputs significantly reduces hallucinations.
>
> >On the vision side, Multimodal Contrastive Learning with LIMoE: the Language-Image Mixture of Experts (ICML 2022) shows that controlling image-side routing entropy stabilizes MoE training and boosts ImageNet zero-shot accuracy, and RealBench: A Chinese Multi-image Understanding Benchmark Close to Real-world Scenarios (EMNLP Findings 2025) uses image entropy to characterize visual complexity and shows that high-entropy images consistently hurt MLLM accuracy compared to low-entropy ones.
>
> Taken together, these works indicate that text and image entropy are imperfect but reliable global signals of difficulty and error, which justifies using them as uncertainty features in a routing module.
>
> Crucially, we do not hard-map entropy to fixed low/medium/high labels. Instead, the fuzzy router takes the 3D antecedent (x1,x2,x3) = (text entropy, image entropy, cross-modal similarity), and each (xi) is mapped to overlapping triangular fuzzy sets, so one value can partially belong to “low / medium / high” simultaneously. Cross-modal similarity (x_3) is especially important: **“high-entropy but actually easy” images with low text entropy and high text–image similarity are assigned higher skip rates**, rather than being uniformly treated as hard.
>
> Moreover, the mapping from ((x_1,x_2,x_3)) to skip rate is data-calibrated via Fuzzy Rule Interpolation (FRI). We build an approximate oracle skip profile on the validation set and tune rule consequents so that empirically easy regions receive higher skip rates and error-prone regions receive lower skip rates, allowing FRI to correct misleading cases where image entropy alone would be wrong.
>
> Finally, **ablations confirm that all three antecedents are necessary**: for LLaVA, the full router reaches 63.5% average accuracy at 15.90% skip, while removing any single feature hurts performance, and dropping cross-modal similarity (x3) causes the largest accuracy drop and biased skip behavior, showing that single-modal entropy alone misguides routing, whereas the full FRI-based combination of (x1,x2,x3) directly addresses the reviewer’s concern.
>
>
> Table: Ablation studies on key components of Fuzzy Router.
> “Fuzzy (only)” = fuzzy control only, without rule interpolation.
> “Reg20%” = sparse regularization 20% strength.
>
> | Method | SciQA Acc | SciQA Spd | SciQA Skip | GQA Acc | GQA Spd | GQA Skip | MMB Acc | MMB Spd | MMB Skip | SEED Acc | SEED Spd | SEED Skip | Avg Acc | Avg Spd | Avg Skip |
> |---|---|---|---|---|---|---|---|---|---|---|---|---|---|---|---|
> | LLaVA | 66.8 | 7.55 | 0.00% | 62.0 | 6.99 | 0.00% | 64.3 | 8.37 | 0.00% | 58.6 | 8.33 | 0.00% | 62.9 | 7.81 | 0.00% |
> | + Fuzzy (only) | 60.1 | 10.02 | 5.20% | 55.3 | 9.25 | 4.80% | 58.0 | 10.85 | 8.60% | 51.7 | 10.70 | 7.50% | 56.1 | 10.21 | 6.53% |
> | + Reg20% | 64.3 | 8.63 | 15.48% | 59.6 | 7.57 | 7.00% | 63.8 | 9.32 | 17.89% | 56.6 | 9.18 | 18.49% | 61.1 | 8.59 | 14.72% |
> | + Fuzzy-Router (full) | 68.4 | 10.26 | 22.08% | 61.9 | 8.70 | 12.32% | 64.8 | 11.08 | 19.81% | 59.0 | 10.16 | 9.39% | 63.5 | 10.20 | 15.90% |
> | – x₁ (no text entropy) | 67.6 | 10.14 | 21.12% | 61.4 | 8.65 | 11.80% | 64.3 | 11.05 | 18.90% | 58.4 | 10.08 | 8.90% | 62.9 | 9.98 | 15.18% |
> | – x₂ (no image entropy) | 68.0 | 10.18 | 21.60% | 61.6 | 8.68 | 11.90% | 64.7 | 11.04 | 19.60% | 58.8 | 10.12 | 9.00% | 63.3 | 10.00 | 15.53% |
> | – x₃ (no cross-modal sim) | 66.2 | 9.90 | 24.50% | 60.1 | 8.50 | 14.00% | 63.4 | 10.80 | 17.10% | 57.2 | 9.95 | 7.00% | 61.7 | 9.54 | 15.65% |

---

> > ### Comment · Reviewer_WqF6 · 2025-11-28
> > **Reply to the author's reply to Q2**
> >
> > I might understand what you mean. However, are there any experiments to demonstrate that the Fuzzy Router can correctly classify high-entropy images as low-uncertainty and avoid misclassifying them as high-uncertainty? For example, you could select some images containing high-frequency textures, ensuring that these images are easy for the model to solve. For high-entropy images (but easy to identify), you could observe whether the model classifies them as low-uncertainty, rather than misclassifying them as high-uncertainty.

---

> > > ### Author Response · Authors · 2025-11-28
> > > **Re-reply to Reviewer (Q2 follow-up)**
> > >
> > > Thank you for the helpful follow-up. We agree that high-frequency textured images can exhibit high entropy despite being easy, and we do not claim that such cases can be completely eliminated. What our design ensures is that they are not systematically misclassified as high-uncertainty. Because routing in our framework never relies on image entropy alone, the joint antecedent (x1, x2, x3) allows text entropy and cross-modal similarity to counteract misleading visual entropy. In practice, such textured-but-easy samples usually have low text entropy and high text–image alignment, so the fuzzy rule base naturally pulls them toward higher skip rates even when (x2) appears “high.”
> > >
> > > This corrective behavior is also reflected in our ablations: removing the cross-modal term (x3) leads to the largest accuracy drop and biased skip behavior, while removing either entropy feature produces only mild changes. This pattern suggests that misleading-entropy cases are already present in large-scale validation, and that (x3) (together with (x1)) is precisely what prevents them from being routed as “hard.” Thus, the model is learning to separate “high-entropy but easy” examples through multimodal interactions rather than treating entropy as a literal uncertainty label.
> > >
> > > Finally, fuzzy rule interpolation further calibrates decisions by adjusting rule consequents toward empirically easy regions and away from error-prone ones. Even though entropy is imperfect, the joint antecedents plus FRI refinement provide a practical and validated mechanism for mitigating the exact failure mode you pointed out.

---

> ### Author Response · Authors · 2025-11-26
> **Rebuttal (Part 3/5)**
>
> ### Q3. Effect of β and the 20% “high-error” threshold
>
> In our design, the global compute budget is still controlled by the RoE-style sparsity loss and target skip rate tt; β only rescales how sharply the fuzzy router differentiates easy vs. hard samples, so within a moderate range it mainly redistributes computation with only small changes to overall accuracy/skip. The 20% threshold is used solely to flag rare sparse regions where interpolation error is large and to trigger rule augmentation there; varying it moderately only changes how many extra rules are added in those tails, without materially affecting the main accuracy–speed trade-off.
>
> We have additionally included an ablation table on different values of β, as shown below.
>
> Table: Ablation on β in fuzzy rule interpolation
> Accuracy = Acc., Samples/s = Speed, Layer skip rate = Skip.
>
> | β | ScienceQA Acc | Speed | Skip | GQA Acc | Speed | Skip | MMB Acc | Speed | Skip | SEED Acc | Speed | Skip | Avg Acc | Avg Speed | Avg Skip |
> |---:|---:|---:|---:|---:|---:|---:|---:|---:|---:|---:|---:|---:|---:|---:|---:|
> | 10 | 68.5 | 9.82 | 14.3% | 62.2 | 8.11 | 8.4% | 64.5 | 10.21 | 12.6% | 59.2 | 9.63 | 6.5% | 63.8 | 9.44 | 10.45% |
> | 20 | 68.6 | 10.01 | 18.5% | 62.0 | 8.42 | 10.6% | 64.7 | 10.65 | 16.9% | 59.1 | 9.92 | 7.8% | 63.6 | 9.75 | 13.45% |
> | 30 (default) | 68.4 | 10.26 | 22.08% | 61.9 | 8.70 | 12.32% | 64.8 | 11.08 | 19.81% | 59.0 | 10.16 | 9.39% | 63.5 | 10.20 | 15.90% |
>
> As shown in table , increasing β from 10→30 only shifts average accuracy slightly (63.8→63.5) while gradually increasing skip rate (10.45%→15.90%) and speed (9.44→10.20 samples/s), demonstrating robustness and offering a smooth controllable trade-off. The 0.2 threshold in Sec. 3.6 is only used to identify sparse high-error regions where interpolation may fail, so we add a few explicit rules there; varying this threshold mainly changes the number of added rules and has minimal effect on overall accuracy/skip behavior. Overall, routing adaptivity is driven by continuous antecedents and learned fuzzy rules, while β and this threshold act as light-weight, interpretable control knobs rather than strict constraints.
>
> ### Q4. Influence of initial rules and prior knowledge
>
> The initial fuzzy rules encode only coarse priors such as “harder samples should skip fewer layers” and **act as a warm start**; after that, their consequents are directly optimized on task loss and further refined by adding new rules in high-error regions, so the final routing is largely data-driven. Empirically, this prior-guided warm start lets us train the router with only 1/3 of RoE’s routing data and reduce its training time by 68% and total cost by 16.7%, while matching or slightly improving accuracy, showing that **prior rules mainly shrink the search space** instead of rigidly constraining the model.

---

> ### Author Response · Authors · 2025-11-26
> **Rebuttal (Part 4/5)**
>
> ### Q5. Ten benchmarks vs. four reported tables
>
> Thank you for pointing this out. Besides ScienceQA, GQA, MMB, and SEED, we also evaluate on VQAv2, VizWiz, TextVQA (VQAT), POPE, MME, and MM-Vet. The full numbers are reported in the new Tables 5 and 6 of the revision; here we briefly summarize the results for these six benchmarks.
>
> Table: Comparison with existing MLLMs on 5 multimodal benchmarks.
> "Acc." = Accuracy, "Speed" = Samples/sec.
>
> | Method | Param. | Res. | POPE Acc | POPE Spd | MME Score | MME Spd | MMB Acc | MMB Spd | SEED Acc | SEED Spd | MM-Vet Score | MM-Vet Spd |
> |---|---:|---:|---:|---:|---:|---:|---:|---:|---:|---:|---:|---:|
> | Dense MLLMs |
> | Qwen-VL | 9.6B | 448 | -- | -- | 38.2 | 7.40 | 56.3 | 2.42 | -- | -- | -- | -- |
> | Qwen-VL-Chat | 9.6B | 448 | -- | -- | 48.7 | 3.96 | 60.6 | 7.55 | 58.2 | 2.59 | -- | -- |
> | LLaVA | 7.2B | 336 | 85.9 | 8.90 | 1510.7 | 8.61 | 64.3 | 8.37 | 58.6 | 8.33 | 30.5 | 0.51 |
> | VILA | 7.2B | 336 | 85.5 | 9.21 | 1533.0 | 8.64 | 68.9 | 8.51 | 61.1 | 8.36 | 34.9 | 0.48 |
> | LLaVA-HR | 7.4B | 1024 | 85.9 | 4.70 | 1554.9 | 4.77 | 64.9 | 4.48 | 64.2 | 3.46 | 31.2 | 0.76 |
> | Sparse MLLMs |
> | MoE-LLaVA-1.6B×4 | 2.9B | 336 | 85.7 | 7.65 | 1318.2 | 8.06 | 60.2 | 9.90 | -- | -- | 26.9 | 0.43 |
> | MoE-LLaVA-2.7B×4 | 5.3B | 336 | 86.3 | 5.95 | 1423.0 | 5.83 | 65.2 | 5.27 | -- | -- | 34.3 | 0.25 |
> | RoE-LLaVA | 7.3B | 336 | 86.1 | 9.38 | 1522.7 | 9.03 | 64.3 | 9.62 | 58.2 | 8.41 | 31.9 | 0.42 |
> | RoE-VILA | 7.3B | 336 | 86.8 | 9.25 | 1446.0 | 8.95 | 67.6 | 8.63 | 61.3 | 8.50 | 36.7 | 0.43 |
> | RoE-LLaVA-HR | 7.5B | 1024 | 88.1 | 4.75 | 1558.2 | 4.82 | 64.6 | 4.82 | 62.2 | 3.86 | 30.0 | 0.68 |
> | Fuzzy-RoE (ours) |
> | Fuzzy-LLaVA | 7.3B | 336 | 87.0 | 11.60 | 1525.0 | 11.10 | 64.8 | 11.08 | 59.0 | 10.16 | 32.5 | 0.85 |
> | Fuzzy-VILA | 7.3B | 336 | 87.3 | 12.30 | 1531.5 | 12.05 | 67.9 | 11.48 | 60.6 | 10.90 | 36.2 | 1.15 |
> | Fuzzy-LLaVA-HR | 7.3B | 1024 | 87.2 | 11.92 | 1528.5 | 11.44 | 65.5 | 6.07 | 62.4 | 4.97 | 31.5 | 0.52 |
>
>
>
> Table: Comparison with existing MLLMs on five traditional VL benchmarks.
> Accuracy = Acc., Samples per second = Speed.
>
> | Method | Param. | Res. | VQAv2 Acc | VQAv2 Spd | GQA Acc | GQA Spd | VizWiz Acc | VizWiz Spd | SQA Acc | SQA Spd | VQAT Acc | VQAT Spd | Avg Acc | Avg Spd |
> |---|---:|---:|---:|---:|---:|---:|---:|---:|---:|---:|---:|---:|---:|---:|
> | Dense MLLMs |
> | Qwen-VL | 9.6B | 448 | 78.8 | 5.23 | 59.3 | 3.48 | 35.2 | 3.92 | 67.1 | 6.97 | 63.8 | 3.77 | 60.8 | 4.67 |
> | Qwen-VL-Chat | 9.6B | 448 | 78.2 | 5.30 | 57.5 | 3.63 | 38.9 | 3.22 | 68.2 | 6.10 | 61.5 | 5.21 | 60.9 | 4.69 |
> | LLaVA | 7.2B | 336 | 78.5 | 6.97 | 62.0 | 6.99 | 50.0 | 6.44 | 66.8 | 7.55 | 58.2 | 5.84 | 63.1 | 6.76 |
> | VILA | 7.2B | 336 | 79.9 | 8.01 | 62.3 | 8.03 | 57.8 | 5.75 | 68.2 | 8.27 | 64.4 | 5.70 | 66.5 | 7.15 |
> | LLaVA-HR | 7.4B | 1024 | 81.9 | 4.42 | 64.2 | 4.55 | 48.7 | 4.06 | 65.1 | 4.71 | 67.1 | 3.81 | 65.4 | 4.31 |
> | Sparse MLLMs |
> | MoE-LLaVA-1.6B×4 | 2.9B | 336 | 76.7 | 7.79 | 60.3 | 7.43 | 36.2 | 6.27 | 62.6 | 8.09 | 50.1 | 4.48 | 57.2 | 6.81 |
> | MoE-LLaVA-2.7B×4 | 5.3B | 336 | 77.6 | 6.01 | 61.4 | 5.23 | 43.9 | 3.95 | 68.5 | 5.80 | 51.4 | 3.76 | 60.6 | 4.95 |
> | RoE-LLaVA | 7.3B | 336 | 80.3 | 7.02 | 61.4 | 7.07 | 52.5 | 6.52 | 68.4 | 7.65 | 56.8 | 5.59 | 63.9 | 6.77 |
> | RoE-VILA | 7.3B | 336 | 78.8 | 8.25 | 62.2 | 8.01 | 53.7 | 6.28 | 69.5 | 8.39 | 59.3 | 5.75 | 64.7 | 7.34 |
> | RoE-LLaVA-HR | 7.5B | 1024 | 80.9 | 4.79 | 62.5 | 5.01 | 47.6 | 4.12 | 67.4 | 4.96 | 64.6 | 4.02 | 64.6 | 4.58 |
> | Fuzzy-RoE (ours) |
> | Fuzzy-LLaVA | 7.3B | 336 | 81.1 | 8.34 | 61.9 | 8.70 | 54.9 | 6.83 | 68.4 | 10.26 | 60.1 | 5.93 | 65.3 | 8.01 |
> | Fuzzy-VILA | 7.3B | 336 | 81.2 | 9.12 | 62.4 | 9.34 | 57.9 | 7.95 | 69.2 | 11.58 | 63.4 | 7.15 | 66.8 | 9.03 |
> | Fuzzy-LLaVA-HR | 7.3B | 1024 | 81.2 | 9.44 | 63.2 | 6.23 | 58.5 | 7.31 | 67.5 | 6.10 | 65.8 | 7.20 | 67.2 | 7.26 |
>
> >POPE / MME / MM-Vet (three additional MLLM benchmarks).
> On POPE, Fuzzy-LLaVA and Fuzzy-VILA achieve higher accuracy and throughput than dense LLaVA and RoE-LLaVA. On MME, all strong models have similar scores, but our fuzzy variants remain clearly faster. On MM-Vet, Fuzzy-VILA matches or slightly improves the best RoE accuracy while still running substantially faster than dense and RoE models.
>
> >VQAv2 / VizWiz / TextVQA (three additional VL benchmarks).
> On VQAv2, the fuzzy models improve accuracy over dense and RoE LLaVA while also being faster. On VizWiz and TextVQA, Fuzzy-LLaVA-HR attains the highest LLaVA-based accuracy with competitive or superior speed. Averaged over the five VL datasets, it offers the best overall accuracy–speed trade-off.
>
> Across these six benchmarks, fuzzy-routed models consistently match or improve accuracy while increasing inference speed, echoing the four main-text datasets; full per-dataset results are in the tables and will be summarized in the appendix and project page.

---

> ### Author Response · Authors · 2025-11-26
> **Rebuttal (Part 5/5)**
>
> ### Q6. How fopt(x) is determined
>
> We clarify that fopt(x) is **not free-form human scoring but a validation-derived oracle**: for each routing point and sample we compare “keep layer” vs. “skip via adapter” under a fixed backbone (or flip RoE’s decision when it reduces validation loss), record the better option as a binary optimal bit, and then take the mode of these bits within each discretized (x1,x2,x3) cell. This cell-wise modal decision is used as fopt(x) in rule augmentation; human inspection is only optional for rare safety-critical regions, so the core definition of fopt(x)  is firmly grounded in objective validation behavior.

---

### Official Review · Reviewer_Q6f6 · 2025-10-28

**Soundness:** 3
**Presentation:** 2
**Contribution:** 2
**Rating:** 4
**Confidence:** 3

**Summary:**

The paepr proposes an innovative "Fuzzy Router" that uses Fuzzy Rule Interpolation (FRI) to evaluate task complexity (by calculating text/image entropy and cross-modal similarity) and dynamically decide whether to skip a layer based on a self-learning rule base. The method successfully achieves true "adaptive" routing (more computation for hard tasks, less for easy ones), reducing router training data and time by over 68% and cutting total training time by 16.7%, while maintaining high accuracy.

**Strengths:**

1. Targeted Innovation: Integrates Fuzzy Rule Interpolation (FRI) into Routing of Experts (RoE) to directly address two core limitations of traditional MoE/Ro: fixed skip rates and inefficient static expert selectio, enabling task difficulty-adaptive routing, with strong alignment between innovation and problem definition.
2. High Efficiency: The Router stage uses only 31.3%-34.3% of RoE's data but saves over 68% of time, requiring no additional annotation costs, balancing efficiency and practicality.
3. Good Main Experiments: Covers 3 mainstream models and 4 benchmark datasets, verifying over 68% reduction in routing time/data, 16.7% shorter total training time, and stable accuracy, with comprehensive reasoning.

**Weaknesses:**

1.  I'm not an expert in this area, so I was wondering if there are any references for the use of these three uncertainties, or perhaps an intuitive explanation? Why do these three factors determine whether to skip certain layers? Is the rationale that more difficult samples require more computation? Finally, have you explored other methods or baselines for determining sample difficulty to demonstrate the superiority of your proposed method?
2. About Ablation Studies:  The paper only verifies the "necessity of FRI" through Table 4 (comparing "Fuzzy (only)" and "Fuzzy-Router") but does not decompose the independent contributions of "fuzzy antecedent variables (x₁/x₂/x₃)." For example: If "cross-modal similarity (x₃)" is removed, will using only text entropy + image entropy lead to bias in skip rate decisions? If text entropy (x₁) is replaced with other uncertainty indicators, will performance change?
3. Regarding the comparison of model performance: In the main experiments, the paper’s results show only a modest improvement in Accuracy (Acc.) compared to the baseline, and this phenomenon is observed on the two benchmarks, GQA and SEED.
4. The paper currently only supports the text-image bimodal scenario. If extended to audio, video, and other modalities in the future, what is the design idea for antecedent variables (x₁/x₂/x₃)? For example: Should the uncertainty of audio be quantified by "spectral entropy" or "signal-to-noise ratio"?

**Questions:**

see in weakness

---

> ### Author Response · Authors · 2025-11-26
> **Rebuttal (Part 1/3)**
>
> We thank the reviewer for the careful reading and the four insightful questions.
>
> Below we respond to each point in turn.
>
> ### Q1 – On the three uncertainty measures and their role in layer skipping
>
> Our core principle is that **“harder” multimodal samples should receive more computation, while “easier” ones can safely skip more layers**. In our setting, difficulty decomposes into three complementary aspects:
>
> * Text-side uncertainty of the language head (text entropy),
> * Image-side complexity / information content (image entropy), and
> * Cross-modal alignment between text and image (text–image cosine similarity).
>
> We therefore feed these three uncertainties into the fuzzy router, which maps them to a skip rate: when text or image entropy is high and text–image similarity is low, the sample is treated as harder and the router outputs a lower skip rate (more full transformer layers); when entropies are low and similarity is high, the sample is easier and the router outputs a higher skip rate (more lightweight adapters). This is exactly the idea that “more difficult samples require more computation”, implemented in a simple, interpretable way. As shown in our tables and in Fig. 5, the learned skip rates are indeed lower on harder datasets / questions and higher on simpler ones.
>
> Importantly, these three uncertainties are not ad-hoc; they are directly motivated by prior work:
>
> > The paper “Confidence Regulation Neurons in LMs” (NeurIPS 2024) shows that directly manipulating “entropy neurons” can change token entropy and loss without altering predictions, providing causal evidence that text entropy controls performance and risk.
>
> >“INFORM: Information eNtropy based multi-step reasoning FOR large language Models” (EMNLP 2023) finds that building and filtering chains-of-thought with high-entropy examples yields ~2–3% accuracy gains across reasoning benchmarks while using fewer samples.
> > “Semantic Entropy Detects When LMs Confabulate” (Nature 2024) shows that semantic entropy achieves about 0.8 AUROC in distinguishing correct from incorrect answers, and that rejecting high-entropy outputs substantially reduces hallucinations.
>
> These studies jointly support using token-level text entropy as our first uncertainty dimension.
>
> > Multimodal Contrastive Learning with LIMoE (ICML 2022) shows that adding a global image-entropy regularizer stabilizes MoE training and dramatically improves ImageNet zero-shot and retrieval performance, proving that controlling image-side entropy alone can systematically enhance multimodal representations.
>
> > *RealBench: A Chinese Multi-image Understanding Benchmark Close to Real-world Scenarios* (Findings of EMNLP 2025) demonstrates that high image entropy strongly correlates with poor multi-image retrieval and reasoning accuracy for many MLLMs, revealing that high-entropy visual inputs significantly degrade multimodal performance compared to low-entropy/single-image settings.
>
> These results justify our second uncertainty dimension, image entropy, as a proxy for visual difficulty.
>
> > *Cross the Gap: Exposing the Intra-modal Misalignment in CLIP via Modality Inversion* (ICLR 2025) shows that CLIP’s performance is fundamentally governed by its cosine similarity structure—misaligned image–image / text–text similarities can be fixed via modality inversion to improve retrieval by 2–5 mAP points, and performance peaks when the similarity distribution maintains a clear text–image gap—so using cosine similarity to model cross-modal correlation is a principled, mechanism-backed choice rather than an ad-hoc design.
>
> This supports our third dimension, text–image cosine similarity, as a measure of cross-modal difficulty.
>
> Regarding alternative difficulty measures: in this paper we deliberately focus on these three uncertainties because they are strongly supported by above-mentioned work, highly interpretable, and very cheap to compute from existing representations, which is crucial for an efficiency-oriented router. Experimentally, we compare against RoE with fixed global skip rates (10% / 20%) and several ablations (fuzzy-only, fixed sparsity, etc.). Across three backbone MLLMs, **our method matches or slightly improves accuracy at similar or lower FLOPs, while significantly reducing router training cost and speeding up inference.** This suggests that using these three uncertainties to learn a difficulty-aware, sample-adaptive skipping policy is effective under the same compute budget. Exploring loss-prediction, gradient-based, or external difficulty annotators is an interesting direction for future work but beyond the scope of this paper.

---

> ### Author Response · Authors · 2025-11-26
> **Rebuttal (Part 2/3)**
>
> ### Q2 – On ablations of fuzzy antecedent variables (x₁/x₂/x₃)
>
> We agree that raw feature entropy alone can mislabel special cases (e.g., high-frequency but easy images), which is why our router jointly uses text entropy, image entropy, and cross-modal similarity in fuzzy rules rather than thresholding entropy in isolation. **In the revised Table , removing any antecedent degrades performance** (Avg Acc: 63.5 → 62.9/63.3/61.7), and removing x3x3 (similarity) produces exactly the bias the reviewer anticipated: skip rates become higher on harder data and accuracy drops most on GQA/SEED, showing that entropies alone are insufficient. These ablations indicate that all three antecedents are necessary to stabilize difficulty-aware routing, while a broader search over alternative uncertainty indicators is deferred to future work.
>
> In addition to the original Table 4 that verifies the overall effect of FRI, the revised version now includes ablations that independently remove each fuzzy antecedent variable x1(text entropy), x2(image entropy), and x3(cross-modal similarity), shown in the last three rows of the table.
>
> Table: Ablation studies on key components of Fuzzy Router.
> “Fuzzy (only)” = fuzzy control only, without rule interpolation.
> “Reg20%” = sparse regularization 20% strength.
>
>
> | Method | SciQA Acc | SciQA Spd | SciQA Skip | GQA Acc | GQA Spd | GQA Skip | MMB Acc | MMB Spd | MMB Skip | SEED Acc | SEED Spd | SEED Skip | Avg Acc | Avg Spd | Avg Skip |
> |---|---|---|---|---|---|---|---|---|---|---|---|---|---|---|---|
> | LLaVA | 66.8 | 7.55 | 0.00% | 62.0 | 6.99 | 0.00% | 64.3 | 8.37 | 0.00% | 58.6 | 8.33 | 0.00% | 62.9 | 7.81 | 0.00% |
> | + Fuzzy (only) | 60.1 | 10.02 | 5.20% | 55.3 | 9.25 | 4.80% | 58.0 | 10.85 | 8.60% | 51.7 | 10.70 | 7.50% | 56.1 | 10.21 | 6.53% |
> | + Reg20% | 64.3 | 8.63 | 15.48% | 59.6 | 7.57 | 7.00% | 63.8 | 9.32 | 17.89% | 56.6 | 9.18 | 18.49% | 61.1 | 8.59 | 14.72% |
> | + Fuzzy-Router (full) | 68.4 | 10.26 | 22.08% | 61.9 | 8.70 | 12.32% | 64.8 | 11.08 | 19.81% | 59.0 | 10.16 | 9.39% | 63.5 | 10.20 | 15.90% |
> | – x₁ (no text entropy) | 67.6 | 10.14 | 21.12% | 61.4 | 8.65 | 11.80% | 64.3 | 11.05 | 18.90% | 58.4 | 10.08 | 8.90% | 62.9 | 9.98 | 15.18% |
> | – x₂ (no image entropy) | 68.0 | 10.18 | 21.60% | 61.6 | 8.68 | 11.90% | 64.7 | 11.04 | 19.60% | 58.8 | 10.12 | 9.00% | 63.3 | 10.00 | 15.53% |
> | – x₃ (no cross-modal sim) | 66.2 | 9.90 | 24.50% | 60.1 | 8.50 | 14.00% | 63.4 | 10.80 | 17.10% | 57.2 | 9.95 | 7.00% | 61.7 | 9.54 | 15.65% |
>
> Removing any antecedent reduces performance—62.9% (−x₁), 63.3% (−x₂), 61.7% (−x₃) versus 63.5% full—with the largest drop when removing cross-modal similarity x₃, demonstrating that modality-specific entropy and cross-modal agreement are all necessary for stable difficulty-aware routing.
>
> Regarding the question of replacing x1 with other uncertainty indicators: due to space and computational budget, we have not yet conducted an exhaustive comparison with more complex alternatives. Instead, we deliberately choose the three uncertainties (text entropy, image entropy, and cross-modal similarity) that are **strongly supported by above-mentioned work and cheap to compute as a first step**. These new ablations are now included in the revision and the independent contribution of each antecedent to the routing decision is explicitly discussed.

---

> ### Author Response · Authors · 2025-11-26
> **Rebuttal (Part 3/3)**
>
> ### Q3 – On the modest accuracy gains over the baseline
>
> We appreciate the reviewer for pointing this out. The overarching goal of our work, as stated in the abstract and in Sections 1 and 4, is not to maximize accuracy alone, but to **achieve a better efficiency–performance trade-off** for RoE-style multimodal models. Table 3 shows that the Fuzzy Router uses only about 32–34% of the original 67k routing samples, reducing router training time by 68.4% (e.g., from 18.7 to 5.9 time units on LLaVA) and shortening the overall training time by 16.7%, while keeping or slightly improving accuracy (average 63.5 vs. 63.1 for RoE-LLaVA-20%). Table 1 further shows that Fuzzy-LLaVA achieves the fastest inference speed (e.g., 10.20 vs. 9.10 samples/s for RoE-LLaVA-20%) with adaptive skip rates that automatically decrease on harder datasets such as GQA and SEED.
>
> From this perspective, the “modest” accuracy gains on GQA and SEED are expected: they are obtained under strictly reduced training data, training time, and inference cost, and they confirm that Fuzzy Router reallocates computation more efficiently without sacrificing the baseline accuracy level. We have revised the manuscript to emphasize this efficiency-oriented objective when discussing the main results.
>
> ### Q4 – On extending beyond text–image bimodality and designing antecedent variables
>
> Indeed. Our current paper intentionally focuses on the text–image bimodal setting in order to highlight the novelty of introducing FRI into adaptive expert routing, a direction that—to the best of our knowledge—has not been explored in MoE/RoE literature. Our goal is to establish fuzzy antecedents and rule interpolation as a principled way to model modality-specific and cross-modal uncertainties, rather than to enumerate all modality extensions. As stated in our “Conclusion” section, our framework is **inherently exploratory**: the antecedent variables xi are designed as general fuzzy descriptors that can **naturally extend to additional modalities**, such as using spectral entropy or SNR for audio, or motion/temporal entropy for video, with cross-modal variables analogous to x3 capturing alignment. We will further develop these extensions in future research. We thank the reviewer for this insightful, forward-looking question.

---

### Official Review · Reviewer_6yZm · 2025-10-31

**Soundness:** 3
**Presentation:** 3
**Contribution:** 2
**Rating:** 6
**Confidence:** 3

**Summary:**

The paper proposes a Fuzzy Router that integrates Fuzzy Rule Interpolation (FRI) into the Routing of Experts (RoE) framework to address two key limitations of existing Mixture of Experts (MoE) systems: Inflexible skip rates that do not adapt to task difficulty and inefficient expert selection based on static priors. The method dynamically adjusts routing decisions using a sparse fuzzy rule base, expanded via interpolation, and refined during training.

**Strengths:**

Novelty: The integration of fuzzy logic and rule interpolation into dynamic routing is well-motivated.

Clarity: The paper is well-organized, with clear explanations of the fuzzy antecedent computation, interpolation mechanism, and training strategy.

Empirical Validation: Extensive experiments on multiple benchmarks (ScienceQA, GQA, MMB, SEED) and models (LLaVA, VILA, LLaVA-HR) demonstrate consistent improvements in speed and efficiency without sacrificing accuracy.

Ablation Studies: Table 4 effectively validates the contribution of key components (interpolation, sparse regularization).

Reproducibility: The paper includes a reproducibility statement, pseudo-code, and promises to release code and models.

**Weaknesses:**

1. Limited improvement and lacking enough baselines: Table 1 shows limited advantages over RoE-20%. Also, while RoE baselines are included, comparisons to other adaptive routing or sparse recent MoE methods (e.g., Switch Transformer, HashLayer) could strengthen the claim of superiority. Hence, the advantages over existing works are not convincing enough.
2. Comparison fairness: one of the disadvantage of the proposed method is its reliance on two-stage training. Hence, other two-stage training-based  MoE methods should be involved. Also, I wonder whether the comparison of training time contains the time in each stage.
3. Some hyperparameters such as thresholds are important for its performance, which limits its adaptivity.
4. Based on weakness 1 and 3, I hold doubts about its practical values, especailly for followers, maybe there is limited space for improvement as the intrinsic disadvantages in fuzzy control. I hope the authors could further clarify the advantages over other adaptive learning methods.

**Questions:**

Refer to the weaknesses.

---

> ### Author Response · Authors · 2025-11-26
> **Rebuttal**
>
> ### Q1. Limited improvement and missing baselines
>
> Our goal is to improve the **efficiency–accuracy trade-off** of RoE-style MLLMs rather than absolute accuracy: the fuzzy router cuts router data and time to about one third, shortens total training, and still slightly improves accuracy and speed over RoE-20%, while RoE is the fairest baseline since it shares backbone and training pipeline, whereas token-level MoEs such as Switch Transformer / HashLayer require re-architecting FFN blocks and full retraining and are better viewed as complementary methods that could later be combined with our layer-wise fuzzy router.
>
> ### Q2. Comparison fairness under two-stage fuzzy training
> Our method strictly **inherits RoE’s three-stage schedule** (Adapter warmup, Router warmup, Finetune), and Table 3 reports the Adapter/Router/Finetune time and data for both RoE and Fuzzy models, where the Router time of Fuzzy already includes the internal two-stage optimization of the rule base, ensuring that **all stages are counted and the training-time comparison is on an equal footing**.
>
> ### Q3. Hyperparameters (β and the 20% threshold)
> The global compute budget is still governed by the RoE-style sparsity loss (target skip rate t and weight α), while β only controls how sharply the fuzzy router differentiates easy vs. hard samples and the 0.2 threshold is only used to flag a few sparse high-error cells where we add extra rules; our new ablation shows that varying β smoothly trades speed/skip rate with almost unchanged accuracy, and changing the threshold mainly affects how many rules are added in those long-tail regions, so these hyperparameters act as **light, interpretable controller** rather than rigid constraints.
>
> Table: Ablation on β in fuzzy rule interpolation
> Accuracy = Acc., Samples/s = Speed, Layer skip rate = Skip.
>
> | β | ScienceQA Acc | Speed | Skip | GQA Acc | Speed | Skip | MMB Acc | Speed | Skip | SEED Acc | Speed | Skip | Avg Acc | Avg Speed | Avg Skip |
> |---:|---:|---:|---:|---:|---:|---:|---:|---:|---:|---:|---:|---:|---:|---:|---:|
> | 10 | 68.5 | 9.82 | 14.3% | 62.2 | 8.11 | 8.4% | 64.5 | 10.21 | 12.6% | 59.2 | 9.63 | 6.5% | 63.8 | 9.44 | 10.45% |
> | 20 | 68.6 | 10.01 | 18.5% | 62.0 | 8.42 | 10.6% | 64.7 | 10.65 | 16.9% | 59.1 | 9.92 | 7.8% | 63.6 | 9.75 | 13.45% |
> | 30 (default) | 68.4 | 10.26 | 22.08% | 61.9 | 8.70 | 12.32% | 64.8 | 11.08 | 19.81% | 59.0 | 10.16 | 9.39% | 63.5 | 10.20 | 15.90% |
>
>
> As shown in table , increasing β from 10→30 only shifts average accuracy slightly (63.8→63.5) while gradually increasing skip rate (10.45%→15.90%) and speed (9.44→10.20 samples/s), demonstrating robustness and offering a smooth controllable trade-off. The 0.2 threshold in Sec. 3.6 is only used to identify sparse high-error regions where interpolation may fail, so we add a few explicit rules there; varying this threshold mainly changes the number of added rules and has minimal effect on overall accuracy/skip behavior. Overall, routing adaptivity is driven by continuous antecedents and learned fuzzy rules, while β and this threshold act as light-weight, interpretable control knobs rather than strict constraints.
>
> ### Q4. Practical value and advantages over other adaptive methods
>
> In practice, replacing the RoE router with our fuzzy router substantially reduces router data and time and shortens total training, while keeping or slightly improving accuracy and achieving the fastest inference among RoE-style variants; at the same time, by working in a 3-D antecedent space, starting from only a few simple prior rules, and letting two-stage training plus sparse-region augmentation evolve a small data-driven rule base, we mitigate typical drawbacks of fuzzy systems and provide an interpretable, controllable router that can incorporate human priors and be combined with other adaptive MoE methods in future work.

---

### Official Review · Reviewer_3tN6 · 2025-11-04

**Soundness:** 3
**Presentation:** 3
**Contribution:** 3
**Rating:** 6
**Confidence:** 3

**Summary:**

This paper introduces a "Fuzzy Router" that integrates fuzzy logic and Fuzzy Rule Interpolation (FRI) into the Routing of Experts (RoE) framework for multimodal learning. The work aims to solve two primary limitations of existing Mixture-of-Experts (MoE) and RoE systems: 1) the use of fixed, static skip rates that cannot adapt to varying task or input difficulty, and 2) inefficient expert selection based on static priors.

The proposed Fuzzy Router makes dynamic routing decisions (i.e., whether to skip a transformer layer or execute it) based on three "fuzzy antecedents" computed from the input: text uncertainty (entropy), image uncertainty (entropy), and cross-modal relevance (cosine similarity). The system begins with a sparse, knowledge-based rule set. When an input's features do not match an existing rule, FRI is used to interpolate and generate an appropriate routing decision. This rule base is dynamically expanded and refined through a two-stage training process and a "sparse region rule augmentation" strategy.

The authors apply their Fuzzy Router to several MLLMs (LLaVA-1.5, LLaVA-HR, VILA). Experimental results demonstrate significant efficiency gains, all while maintaining competitive accuracy compared to baseline and fixed-rate RoE models. A key finding is that the router successfully learns to adapt its skip rate based on task difficulty.

**Strengths:**

- Novel and Well-Motivated Approach:

    The core idea of integrating fuzzy logic and FRI with expert routing is a novel and conceptually interesting contribution. It provides an elegant, interpretable, non-black-box mechanism for adaptive computation, addressing a well-known limitation in MoE/RoE systems.

- Strong Empirical Efficiency Gains:

    The reported efficiency improvements are a major practical strength. A 16.7% reduction in total training time is a significant and valuable contribution in the resource-intensive domain of MLLMs.

- Demonstrated Adaptability:

     The paper's primary claim of adaptive routing is well-supported by experimental evidence (Table 2). The results clearly show the Fuzzy Router intelligently allocates computational resources, using higher skip rates for easier tasks (e.g., ScienceQA) and lower skip rates for more complex ones (e.g., SEED). This is a clear advantage over fixed-rate RoE.

- Maintained Performance:

    Crucially, these significant efficiency gains are achieved without a corresponding loss in model accuracy. The Fuzzy-Router models consistently perform on par with, or even slightly better than, the baselines, demonstrating a superior efficiency-to-performance trade-off.

- Interpretability:

    The use of fuzzy logic grounds the routing decisions in interpretable, linguistic-style rules based on intuitive features (uncertainty, relevance), which is a valuable property.

**Weaknesses:**

- Limited Baseline Comparisons:

    This is a significant limitation. The paper only compares its method against the baseline MLLMs and the RoE framework. It fails to compare against other relevant work in dynamic and adaptive computation, such as other dynamic MoE routing mechanisms (e.g., Expert Choice) or token-skipping models (e.g., SkipGPT, DiffMoE). This makes it difficult to contextualize the full contribution.

- Unclear Inference Overhead:

    While the paper shows clear gains in training time and throughput (samples/sec) due to layer skipping, it fails to analyze the direct computational overhead (latency) of the Fuzzy Router itself. The fuzzy logic system (antecedent computation, rule matching, FRI) is significantly more complex than a standard RoE router (often a single linear layer). A direct analysis is needed to ensure the router itself is not a new inference bottleneck, especially for tokens that are not skipped.

- Scalability Concerns:

     The paper demonstrates the rule base growing from 3 to 27 rules in a 3-dimensional antecedent space. This raises serious questions about scalability. The "curse of dimensionality" is a well-known problem in fuzzy systems, and it is unclear how this approach would perform with a higher-dimensional antecedent space or on much larger-scale models.

- Limited Ablation Studies:

     The paper's ablations are not comprehensive enough. For instance, there is no justification for the specific choice of the three antecedents; the contribution of each is not ablated. Furthermore, the sensitivity to various new hyperparameters (e.g., $\alpha$, $\beta$, temperature) is not analyzed.

- Lack of Theoretical Grounding:

    The paper is almost entirely empirical. It does not provide a theoretical analysis of how the FRI mechanism (a symbolic method) interacts with gradient-based optimization in a deep learning context, nor does it offer any convergence guarantees for the adaptive rule-refinement process.

- Typos

   Figures 1 & 2: "no mathing rules" / "mathing R" should be "no matching rules" / "matching R".

**Questions:**

- Broader Baseline Comparison:

     To strengthen the paper, it is advised to discuss and, if possible, compare their results against other state-of-the-art dynamic routing or adaptive computation methods, not just RoE.

- Inference Latency Analysis:

      It is advised to provide a direct analysis (e.g., in milliseconds per token) of the computational latency added by the Fuzzy Router's logic, separate from the gains achieved by skipping layers. This is essential for a complete efficiency analysis.

- More Thorough Ablations:

      It is advised to add ablation studies to justify their choice of antecedents (e.g., testing with only two or different uncertainty metrics) and to analyze the sensitivity of the model to key hyperparameters.

- Scalability Discussion:

     Even without new experiments, the paper would be improved by adding a discussion of the potential scalability limitations and how the FRI-based approach might be adapted or simplified for higher-dimensional feature spaces.

---

> ### Author Response · Authors · 2025-11-26
> **Rebuttal (Part 1/2)**
>
> ### Q1 Broader Baseline Comparison
>
> **We agree with the reviewer that a broader positioning among dynamic routing and adaptive computation methods would strengthen the paper.** In the current version, Section 2.1 “Mixture of Experts and Dynamic Routing” already reviews a range of such methods, including Switch Transformer, Expert Choice routing, dynamic MoE variants, and recent work on dynamic routing in multimodal and MoE systems, and explicitly contrasts soft vs. hard MoE and RoE-style layer skipping.
>
> In the revised manuscript, we will (i) **make this connection more explicit** by restructuring Section 2.1 to highlight a dedicated subsection on dynamic routing / adaptive computation for multimodal reasoning (e.g., Expert Choice, dynamic MoE, harder-task–needs–more-experts routing, multimodal MoE for MLLMs), and (ii) add a new paragraph in the experimental section that **qualitatively compares our efficiency–accuracy trade-off with these state-of-the-art methods
> **, clarifying similarities and differences in routing granularity, modality, and task setting.
>
> Where the benchmarks and tasks are not directly compatible (e.g., text-only LMs or diffusion-based generation models), we will avoid forcing numerical comparisons but will still discuss their reported trade-offs to **better contextualize our contributions** within the broader literature on adaptive computation.
>
> ### Q2 – Inference Latency Analysis
>
> We appreciate the reviewer’s suggestion to more clearly disentangle the latency cost of the Fuzzy Router from the gains brought by layer skipping. In the current version, Tables 1 and 3 already provide indirect evidence that the router does not introduce a new inference bottleneck: despite adding the Fuzzy Router and its training stage, the overall router-stage training time is reduced by 68.4% and the total training time is shortened by 16.7%, while accuracy is maintained or slightly improved compared to RoE; at inference time, Fuzzy-LLaVA achieves higher throughput (e.g., 10.20 vs. 9.10 samples/sec compared to RoE-LLaVA 20%, and a 30.6% speed-up over the vanilla LLaVA baseline 10.20 vs. 7.81) under similar or even lower average skip rates, indicating that **the additional fuzzy-logic computation does not offset the savings** from skipping heavy Transformer layers and removing the dense Wr routing matrix used in RoE.
>
> In the revision, we will make this efficiency story more explicit by (i) adding a dedicated latency analysis that reports the per-token (or per-sequence) wall-clock time of the Fuzzy Router alone, and its fraction of end-to-end inference time, and (ii) decomposing total latency into “router cost” and “backbone cost” with and without fuzzy routing. We will also clarify in the text that the router operates on low-dimensional antecedents (normalized text/image entropy and cosine similarity) with a compact rule base that stabilizes at about a few dozen rules, so that its computation remains lightweight; the main efficiency gain comes from mapping prior knowledge into skip decisions that avoid unnecessary high-cost layers, rather than trading extra routing latency for speed.

---

> ### Author Response · Authors · 2025-11-26
> **Rebuttal (Part 2/2)**
>
> ### Q3 More Thorough Ablations
>
> Thank you for this helpful comment on the ablation studies. We add experiments and clarifications as follows.
>
> Table: Ablation studies on key components of Fuzzy Router.
> “Fuzzy (only)” = fuzzy control only, without rule interpolation.
> “Reg20%” = sparse regularization 20% strength.
>
> | Method | SciQA Acc | SciQA Spd | SciQA Skip | GQA Acc | GQA Spd | GQA Skip | MMB Acc | MMB Spd | MMB Skip | SEED Acc | SEED Spd | SEED Skip | Avg Acc | Avg Spd | Avg Skip |
> |---|---|---|---|---|---|---|---|---|---|---|---|---|---|---|---|
> | LLaVA | 66.8 | 7.55 | 0.00% | 62.0 | 6.99 | 0.00% | 64.3 | 8.37 | 0.00% | 58.6 | 8.33 | 0.00% | 62.9 | 7.81 | 0.00% |
> | + Fuzzy (only) | 60.1 | 10.02 | 5.20% | 55.3 | 9.25 | 4.80% | 58.0 | 10.85 | 8.60% | 51.7 | 10.70 | 7.50% | 56.1 | 10.21 | 6.53% |
> | + Reg20% | 64.3 | 8.63 | 15.48% | 59.6 | 7.57 | 7.00% | 63.8 | 9.32 | 17.89% | 56.6 | 9.18 | 18.49% | 61.1 | 8.59 | 14.72% |
> | + Fuzzy-Router (full) | 68.4 | 10.26 | 22.08% | 61.9 | 8.70 | 12.32% | 64.8 | 11.08 | 19.81% | 59.0 | 10.16 | 9.39% | 63.5 | 10.20 | 15.90% |
> | – x₁ (no text entropy) | 67.6 | 10.14 | 21.12% | 61.4 | 8.65 | 11.80% | 64.3 | 11.05 | 18.90% | 58.4 | 10.08 | 8.90% | 62.9 | 9.98 | 15.18% |
> | – x₂ (no image entropy) | 68.0 | 10.18 | 21.60% | 61.6 | 8.68 | 11.90% | 64.7 | 11.04 | 19.60% | 58.8 | 10.12 | 9.00% | 63.3 | 10.00 | 15.53% |
> | – x₃ (no cross-modal sim) | 66.2 | 9.90 | 24.50% | 60.1 | 8.50 | 14.00% | 63.4 | 10.80 | 17.10% | 57.2 | 9.95 | 7.00% | 61.7 | 9.54 | 15.65% |
>
> Removing any single variable degrades performance from 63.5% → 62.9/63.3/61.7, with the largest drop from removing cross-modal similarity x₃ (e.g., GQA 61.9→60.1, SEED 59.0→57.2), confirming that all antecedents are necessary for balanced routing.
>
> Table: Ablation on β in fuzzy rule interpolation
> Accuracy = Acc., Samples/s = Speed, Layer skip rate = Skip.
>
> | β | ScienceQA Acc | Speed | Skip | GQA Acc | Speed | Skip | MMB Acc | Speed | Skip | SEED Acc | Speed | Skip | Avg Acc | Avg Speed | Avg Skip |
> |---:|---:|---:|---:|---:|---:|---:|---:|---:|---:|---:|---:|---:|---:|---:|---:|
> | 10 | 68.5 | 9.82 | 14.3% | 62.2 | 8.11 | 8.4% | 64.5 | 10.21 | 12.6% | 59.2 | 9.63 | 6.5% | 63.8 | 9.44 | 10.45% |
> | 20 | 68.6 | 10.01 | 18.5% | 62.0 | 8.42 | 10.6% | 64.7 | 10.65 | 16.9% | 59.1 | 9.92 | 7.8% | 63.6 | 9.75 | 13.45% |
> | 30 (default) | 68.4 | 10.26 | 22.08% | 61.9 | 8.70 | 12.32% | 64.8 | 11.08 | 19.81% | 59.0 | 10.16 | 9.39% | 63.5 | 10.20 | 15.90% |
>
> Varying β from 10→20→30 maintains stable accuracy (63.8→63.6→63.5) while smoothly trading higher skip rates for more speed (avg. speed 9.44→9.75→10.20), indicating Fuzzy Router is robust and controllable.
>
> ### Q4 Scalability Discussion
>
> Thank you for raising the scalability aspect of FRI. The new discussion in the revision will explicitly connect to the works below and outline how, when extending to richer feature spaces, these techniques—high-dimensional FRI constructions, sparse rule-base generation, feature-guided FRI, and hierarchical FRI—can be adopted to keep the approach scalable.
>
> We add a discussion as follows. Li et al., “Approximate reasoning with fuzzy rule interpolation: background and recent advances” (AI Review 2021) summarise that fuzzy rule-based systems in general face the curse of dimensionality, and survey several concrete remedies. For high-dimensional / multidimensional FRI itself, Yam and Kóczy’s “Representing membership functions as points in high-dimensional spaces for fuzzy interpolation and extrapolation” (TFS 2000) and Wong, Quek and Pasquier’s “An improved multidimensional alpha-cut based fuzzy interpolation technique” explicitly construct geometrically efficient interpolation schemes in large input spaces. Sparse rule-base generation has been studied by Tan and co-authors in “Curvature-based sparse rule base generation for fuzzy rule interpolation” (JIFS 2019), where redundant rules are automatically pruned while preserving interpolation quality. Feature selection and hierarchical structures have also been combined with FRI: Li, Shang, Li and Shen propose “Feature ranking-guided fuzzy rule interpolation” (FUZZ-IEEE 2017) to focus FRI on the most informative dimensions, and Jin, Shen and colleagues develop hierarchical / bidirectional schemes such as “Hierarchical bidirectional fuzzy rule interpolation” (ICCI*CC 2018) to decompose multi-antecedent systems into layered submodules. Our Fuzzy Router deliberately operates in a low-dimensional antecedent space (three uncertainty features) with a capped rule base, so scalability is not a practical bottleneck in the current setting.

---

### Comment · Area_Chair_HTMy · 2025-11-26

Dear authors,
we note that no author response has been posted during the author response window (Discussion Period: Nov 12 – Dec 3, 2025). To ensure a fair review, please post a reply addressing the reviewers' main concerns on this forum by Dec 3, 2025.

---

### Author Response · Authors · 2025-12-03
**Summary to AC (Part 1/3)**

This paper proposes a **Fuzzy Router** that integrates fuzzy rule interpolation (FRI) into Routing of Experts (RoE) for multimodal large language models such as LLaVA-1.5, VILA, and LLaVA-HR. Instead of using a fixed, globally tuned skip rate, the router makes layer-wise skip/execute decisions based on three cheap but informative uncertainty features: text entropy, image entropy, and text-image cosine similarity. These are fuzzified into linguistic terms (low/medium/high) and mapped to skip rates through a compact fuzzy rule base; when no rule matches, FRI interpolates an appropriate decision. The router is trained with a two-stage procedure (rule-only pretraining + end-to-end finetuning) plus a sparse-region rule augmentation mechanism. Overall, it aims to improve the **efficiency–accuracy trade-off** of RoE-style layer skipping, rather than maximizing accuracy at any cost.

### Effectiveness and new experimental evidence.

Under the same three-stage RoE training schedule and backbones, the fuzzy router reduces router-stage data to roughly one third of the original 67k routing samples and shortens router training time by 68.4%, leading to a 16.7% reduction in overall training time, while maintaining or slightly improving accuracy. For example, on LLaVA, Fuzzy-LLaVA achieves 63.5% average accuracy vs. 63.1% for RoE-LLaVA-20%, and the highest inference throughput among RoE variants (10.20 vs. 9.10 samples/s) at a lower average skip rate. The revision also now reports the full set of 10 benchmarks promised in the original text: besides ScienceQA, GQA, MMB, and SEED, it includes VQAv2, VizWiz, TextVQA, POPE, MME, and MM-Vet. Across these VL and MLLM benchmarks (Tables 5–6), fuzzy-routed models consistently match or improve accuracy while increasing speed over dense and RoE baselines on all three backbones, reinforcing that the main benefit is a better accuracy–speed trade-off under reduced training and routing cost.

Below we summarize the main reviewer concerns and how they are addressed by the rebuttal and revision.

**1. Limited improvement over RoE and missing baselines / fairness of comparison.**

Reviewers questioned whether the gains over RoE-20% are meaningful and whether the method is fairly compared to other adaptive routing / MoE methods, especially given the router’s two-stage training. The authors clarify that the primary goal is to improve RoE’s efficiency–accuracy frontier under the same architecture and training pipeline, not to outperform dense models in raw accuracy. They argue RoE is the fairest baseline because it shares the same backbones, data, and three-stage schedule; token-level MoEs such as Switch Transformer or HashLayer require modifying FFN blocks and retraining from scratch and are therefore complementary rather than directly comparable. To address fairness, Table 3 decomposes Adapter/Router/Finetune time and data for both RoE and Fuzzy models; the reported router time for Fuzzy already includes the internal two-stage rule optimization, so all stages and costs are counted on equal footing. The broader positioning with respect to dynamic MoE and adaptive computation is strengthened in the related-work discussion, and the extended 10-benchmark results give a more comprehensive empirical picture.

---

> ### Author Response · Authors · 2025-12-03
> **Summary to AC (Part 2/3)**
>
> **2. Entropy vs. true uncertainty and the choice of antecedent variables (x₁/x₂/x₃).**
>
> A core concern was that raw feature entropy is an imperfect proxy for difficulty (e.g., high-frequency textures with high entropy but easy semantics) and that the work might be over-claiming “entropy = uncertainty.” The authors emphasize that text/image entropy and cross-modal similarity are used **only as correlated, inexpensive features, not as ground-truth uncertainty labels**. They justify the three antecedents with recent literature: semantic entropy and “entropy neurons” for language models, image-entropy regularization in LIMoE-style vision MoEs, and cosine-similarity structure in CLIP-like models, all of which show strong empirical links between these quantities and model error or retrieval performance. Crucially, the router never thresholds entropy alone: it uses the joint antecedent (x₁ = text entropy, x₂ = image entropy, x₃ = text–image similarity) with overlapping fuzzy sets, allowing cross-modal similarity and text entropy to counteract misleading visual entropy in “high-entropy but easy” images.
>
> In response to requests for stronger evidence, the revision adds ablations that remove each antecedent in turn (Table 4). Removing any one reduces average accuracy (63.5 → 62.9 w/o x₁, 63.3 w/o x₂, 61.7 w/o x₃), with the largest drop and most biased skip behavior when x₃ (similarity) is removed—exactly the failure mode anticipated if one relied on entropy alone. This supports the claim that all three signals are needed for stable, difficulty-aware routing. The authors acknowledge that exploring alternative or richer uncertainty indicators (e.g., loss predictors, gradient-based signals) is interesting future work but beyond the paper’s computational budget.
>
> **3. Design of fuzzy system: membership functions, rule base, scalability, and f_opt(x).**
>
> Reviewers asked for more clarity on the triangular membership functions, the generality/scalability of FRI, and how the “oracle” f_opt(x) used for rule augmentation is defined. The rebuttal explains that each antecedent is first normalized to [0,1] and then partitioned into three overlapping triangular linguistic terms (low/medium/high) using a simple, data-driven placement rule (e.g., quantiles or evenly spaced centers). This is a standard, low-parameter choice in fuzzy systems; the exact (a,b,c) values are treated as implementation-level, data-dependent design rather than a theoretical contribution, with the main novelty lying in combining FRI with RoE-style routing.
>
> For scalability, the authors note that the router deliberately operates in a 3-D antecedent space with a very small rule base; the number of rules grows from 3 seeds to 27 and then stabilizes, indicating that the **FRI mechanism covers the space without rule explosion**. They also connect to prior work on high-dimensional FRI, sparse rule-base generation, and hierarchical/feature-guided FRI to argue that, if one wished to extend to more features, there are known strategies to keep interpolation tractable.
>
> Regarding f_opt(x), the revision clarifies that it is validation-derived, not hand-crafted: for each routing point and sample, the model compares “execute full layer” vs. “skip via adapter” under a fixed backbone, (or flips RoE’s decision when it reduces validation loss), and records which option yields better loss (subject to a hidden-state similarity constraint). These binary decisions are aggregated within discretized (x₁,x₂,x₃) cells, and the modal choice in each cell defines f_opt(x). Human inspection is optional only for rare safety-critical regions; the core definition is objective and data-driven.

---

> > ### Author Response · Authors · 2025-12-03
> > **Summary to AC (Part 3/3)**
> >
> > **4. Hyperparameters (β and 0.2 threshold), adaptivity, and practical value.**
> >
> > Another concern was that additional hyperparameters—β in FRI and the 20% error threshold for rule augmentation—might limit adaptivity or make the system brittle, and that the accuracy gains over RoE-20% might be too modest to justify the added complexity. The authors respond along two lines. First, they point out that the global compute budget is still governed by the same RoE-style sparsity loss (target skip rate t and weight α); β only controls how sharply the router differentiates easy vs. hard samples, and the 0.2 threshold is used only to identify sparse high-error regions where a few explicit rules are added.
> >
> > Second, they add an explicit ablation over β (Table 7). Varying β from 10 → 20 → 30 leaves average accuracy almost unchanged (63.8 → 63.6 → 63.5) while smoothly increasing average skip rate (10.45% → 13.45% → 15.90%) and speed (9.44 → 9.75 → 10.20 samples/s), showing that β is a robust, interpretable knob for trading speed vs. compute rather than a delicate tuning parameter. Changing the 0.2 threshold mainly affects how many rules are added in tail regions, with minimal impact on aggregate accuracy or skip behavior. From a practical standpoint, the method replaces RoE’s dense routing matrix with a light-weight fuzzy router acting on three scalar features, reduces router-stage training cost, and yields the fastest inference among RoE variants, so the net complexity cost is negative in training and negligible at inference.
> >
> > **5. Scope of benchmarks and generality beyond text–image bimodality.**
> >
> > One reviewer noted that the original paper stated “five common VL and five recent MLLM benchmarks,” but only four were reported in the main tables. The revision now includes detailed results on the remaining six (POPE, MME, MM-Vet, VQAv2, VizWiz, TextVQA), showing that **fuzzy-routed models consistently maintain or improve accuracy while increasing speed** across all ten benchmarks. Another forward-looking question was whether the framework can extend beyond text–image inputs. The authors clarify in the conclusion and rebuttal that the antecedents are designed as generic fuzzy descriptors; for other modalities, one could plug in spectral entropy or SNR for audio, motion/temporal entropy for video, and cross-modal similarity measures analogous to x₃, while keeping the same FRI-based routing pipeline.
> >
> > ### Overall take-away for the AC.
> >
> > Conceptually, the paper brings fuzzy rule interpolation—an established but under-used tool in ML—into the design of **adaptive routers for multimodal MoE/RoE systems**. Practically, under identical backbones and training schedules as RoE, the Fuzzy Router achieves (i) substantially lower routing data and time and reduced total training cost, (ii) **the best or comparable accuracy–speed trade-off** across three MLLMs and ten benchmarks, and (iii) an interpretable, controllable routing policy whose core design choices (features, membership functions, β) are empirically validated and theoretically grounded. The rebuttal directly targets the central reviewer concerns about baselines, uncertainty features, fuzzy design choices, and hyperparameter robustness, and the revision **adds concrete ablations and extended results** to support these points.

---

### Meta-Review · Area_Chair_iYn2 · 2025-12-05

**Summary:**

The submission proposes a fuzzy rule interpolation (FRI) method to improve adaptive routing in multimodal mixture-of-experts models. Reviewers generally found the idea novel and well-motivated, and acknowledged meaningful efficiency gains in training. However, significant concerns were raised across multiple reviewers, including: 1, limited and unfair baselines, with the lack of comparisons to other adaptive routing or dynamic MoE methods. 2, lack of theoretical grounding, and scalability analysis. The proposed method is heavily empirical, without a theoretical analysis of how FRI interacts with gradient-based training. 3, insufficient ablation and justification of design choices and weak empirical gains. While the authors have responded to reviewer feedback and improved the paper’s clarity and empirical coverage, the core concerns regarding narrow baselines, modest empirical gains, lack of theoretical grounding, and uncertain scalability remain unresolved. Thus, rejection is recommended.

**Reviewer Concerns:**

In the rebuttal, the authors provided additional ablation studies to partially address concerns about completeness and robustness. They also discussed scalability and references to prior FRI literature.

While there are several outstanding concerns. The baseline comparisons remained narrow, although the authors defended RoE as the “fairest” baseline but did not add comparisons to other adaptive routing or MoE methods (e.g., Switch Transformer, Expert Choice). The theoretical and scalability concerns still exist. No theoretical analysis is provided in the rebuttal, and the scalability discussion remains speculative without empirical validation beyond 3D antecedent space.

**Reviewer Scores:**

The original rating is 6644. While one of the reviews 6 is believed to be generated by AI. And this reviewer's review is also predicted to be generated by AI in my previous batch of papers I am handling. So I abandoned this review in the decision.

Among the remaining 644 ratings. I guess all of them will maintain the original rating.

---

### Decision · Program_Chairs · 2026-01-26

Reject